# Genome-wide association meta-analysis yields 20 loci associated with gallstone disease

Egil Ferkingstad [1], Asmundur Oddsson [1], Solveig Gretarsdottir[1], Stefania Benonisdottir[1],
Gudmar Thorleifsson[1], Aimee M. Deaton[1], Stefan Jonsson[1], Olafur A. Stefansson[1], Gudmundur L. Norddahl[1],
Florian Zink[1], Gudny A. Arnadottir [1], Bjarni Gunnarsson[1], Gisli H. Halldorsson [1], Anna Helgadottir [1],
Brynjar O. Jensson[1], Ragnar P. Kristjansson[1], Gardar Sveinbjornsson[1], David A. Sverrisson[1], Gisli Masson[1],
Isleifur Olafsson[2], Gudmundur I. Eyjolfsson[3], Olof Sigurdardottir[4], Hilma Holm[1], Ingileif Jonsdottir[1,5,6],
Sigurdur Olafsson[7], Thora Steingrimsdottir[5,8], Thorunn Rafnar [1], Einar S. Bjornsson[5,7],
Unnur Thorsteinsdottir[1,5], Daniel F. Gudbjartsson [1,9], Patrick Sulem [1] & Kari Stefansson [1,5]

Gallstones are responsible for one of the most common diseases in the Western world and are commonly treated with cholecystectomy. We perform a meta-analysis of two genome-wide association studies of gallstone disease in Iceland and the UK, totaling 27,174 cases and 736,838 controls, uncovering 21 novel gallstone-associated variants at 20 loci. Two distinct low frequency missense variants in *SLC10A2*, encoding the apical sodium-dependent bile acid transporter (ASBT), associate with an increased risk of gallstone disease (Pro290Ser: OR = 1.36 [1.25–1.49], $P = 2.1 \times 10^{-12}$, MAF = 1%; Val98Ile: OR = 1.15 [1.10–1.20], $P = 1.8 \times 10^{-10}$, MAF = 4%). We demonstrate that lower bile acid transport by ASBT is accompanied by greater risk of gallstone disease and highlight the role of the intestinal compartment of the enterohepatic circulation of bile acids in gallstone disease susceptibility. Additionally, two low frequency missense variants in *SERPINA1* and *HNF4A* and 17 common variants represent novel associations with gallstone disease.

[1] deCODE Genetics/Amgen, Inc., Reykjavik 101, Iceland. [2] Department of Clinical Biochemistry, Landspítali University Hospital, Reykjavik 101, Iceland. [3] Laboratory in Mjódd (RAM), Reykjavik 109, Iceland. [4] Department of Clinical Biochemistry, Akureyri Hospital, Akureyri 600, Iceland. [5] Faculty of Medicine, University of Iceland, Reykjavik 101, Iceland. [6] Department of Immunology, Landspitali University Hospital, Reykjavik 101, Iceland. [7] Department of Internal Medicine, Landspitali University Hospital, Reykjavik 101, Iceland. [8] Department of Obstetrics and Gynecology, Landspitali University Hospital, Reykjavik 101, Iceland. [9] School of Engineering and Natural Sciences, University of Iceland, Reykjavik 101, Iceland. These authors contributed equally: Egil Ferkingstad, Asmundur Oddsson. Correspondence and requests for materials should be addressed to P.S. (email: patrick.sulem@decode.is) or to K.S. (email: kstefans@decode.is)

Gallstone disease (also known as cholelithiasis) is a common condition affecting 10–15% of the population in developed countries[1], and is often complicated by cholecystitis, pancreatitis and secondary infections[2]. Gallstones are masses of solid material formed in the gallbladder, most commonly consisting of cholesterol[2]. Excess of cholesterol or a lack of bile acids increase the likelihood of gallstone formation[3]. Treatment by surgical removal of the gallbladder (cholecystectomy) is one of the most common surgeries in the Western world, with over 700,000 cholecystectomies performed annually in the United States[1].

A twin study estimated that genetic factors account for 25% [95% CI: 9–40%] of the phenotypic variation of gallstone disease[4]. Sequence variants at eight loci have been reported through genome-wide association studies (GWAS) to affect the risk of gallstone disease, including missense variants in *ABCG5/ABCG8*[5], encoding ATP-binding cassette (ABC) transporters involved in the secretion of cholesterol into bile[6]. We have reported two rare coding variants in *ABCB4* that greatly increase the risk of gallstone disease[7]. *ABCB4* encodes the ABC transporter responsible for the secretion of lecithin into bile[6]. Other studies have reported sequence variants at *GCKR*[8], *TM4SF4*[8], *SULT2A1*[8], *CYP7A1*[8], *UGT1A6*[9], and *TTC39B*[10] to associate with risk of gallstone disease.

To search for sequence variants affecting the risk of gallstone disease, we perform a genome-wide association study (GWAS) of 27,174 cases and 736,838 controls from Iceland and the UK biobank. Subsequently, we assess the association of gallstone disease variants with acute pancreatitis, cholecystitis, fibrosis/cirrhosis of liver, cholestasis of pregnancy, gallbladder cancer, six liver biomarkers, and three cholesterol traits. We find four novel low-frequency sequence variants in *SLC10A2* (two distinct variants), *SERPINA1* and *HNF4A* that associate with gallstone disease. *SLC10A2* encodes a protein that reabsorbs bile salts from the terminal ileum as part of their enterohepatic circulation. Additionally, we describe 17 novel common variants that associate with gallstone disease. Among those, two are in fucosyltransferase genes, four associate with maturity onset diabetes of the young, and six associate with lipid metabolism. We systematically assess the effects of lipid associated variants with gallstone disease risk and find inconsistent direction of effect. With the discovery of gallstone-associated variants in *SLC10A2*, we highlight a role of the intestinal compartment of the enterohepatic circulation of bile acids in gallstone disease susceptibility.

## Results

**Overview.** We performed a meta-analysis of gallstone disease, combining GWAS results from Iceland (8757 cases and 346,688 controls) and the UK Biobank (18,417 cases and 390,150 controls), for a total of 27,174 cases and 736,838 controls (Fig. 1, Supplementary Table 1). The GWAS from Iceland was performed using 32 million markers identified through whole-genome sequencing of 15,220 Icelanders that were subsequently imputed into 151,677 chip-typed individuals, as well as 282,894 of their first and second degree relatives[7]. The GWAS from the UK was performed with 40 million markers (imputation info > 0.8) from the Haplotype Reference Consortium (HRC) reference panel, imputed into chip-typed individuals of European ancestry in the UK Biobank[11] (Fig. 1). Associations were considered significant if the p-value in the combined dataset was below a weighted genome-wide significance threshold based on variant annotation[12]. The significance thresholds used were $2.0 \times 10^{-7}$ for high-impact variants (including stop-gained, frameshift, splice acceptor or donor, $N = 11,465$), $3.9 \times 10^{-8}$ for moderate-impact variants (including missense, splice-region variants and in-frame indels, $N = 197,583$), $3.6 \times 10^{-9}$ for low-impact variants (including upstream and downstream variants, $N = 2,971,445$) and $5.9 \times 10^{-10}$ for lowest-impact variants (including intron and intergenic variants, $N = 39,726,619$). In total, 43 million markers were tested in the meta-analysis.

We found 32 gallstone disease association signals at 28 loci (Table 1, Supplementary Table 2, Fig. 2, and Supplementary Data 1) that reached genome-wide significance under the additive model. Of these, 21 are novel and 11 correspond to previously known variants[5,8,9,13]. One of the novel common variants, rs708686 upstream of *FUT6*, associates more significantly with gallstone disease through a recessive mode of inheritance. There was no significant heterogeneity of effects between Iceland and the UK (Table 1, Supplementary Data 1). Thirteen of the novel signals represent associations with coding variants, four of which are low frequency coding variants, with minor allele frequency (MAF) ranging from 0.6 to 4.6% (imputation info > 0.995 in both Iceland

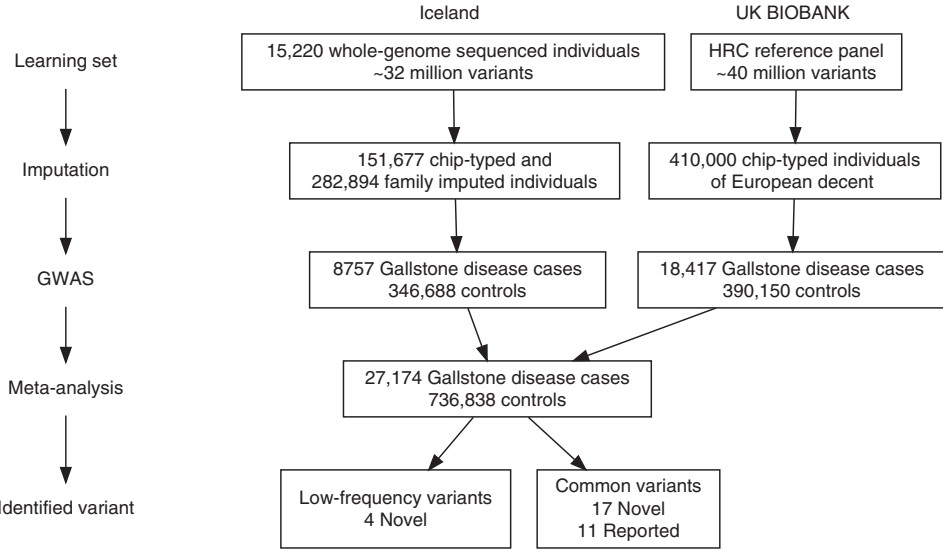

**Fig. 1** A flowchart describing the study design of the gallstone meta-analysis of Icelandic and UK Biobank data and results

**Table 1 Novel gallstone disease associated variants found in the meta-analysis of Icelandic and UK Biobank data**

| Marker | Position (Hg38) | min/maj | MAF (%) | Variant | Gene | LD-class (HIGH/MOD/LOW/LOWEST) | OR (95% CI) | P-value | $P_{het}$ | Functional support |
|---|---|---|---|---|---|---|---|---|---|---|
| rs1800961 | chr20:44413724 | T/C | 4.6 | Thr139Ile | HNF4A | 1 (0/1/0/0) | 1.29 [1.23,1.35] | $5.7 \times 10^{-26}$ | 0.23 | Master regulator of liver specific genes |
| rs28929474* | chr14:94378610 | T/C | 0.81 | Glu366Lys | SERPINA1 | 2 (0/1/1/0) | 1.33 [1.25,1.42] | $1.8 \times 10^{-17}$ | 0.27 | OMIM:613490 alpha-1-antitrypsin deficiency |
| rs56398830 | chr13:103049340 | A/G | 0.59 | Pro290Ser | SLC10A2 | 2 (0/1/0/1) | 1.36 [1.25,1.49] | $2.1 \times 10^{-12}$ | 0.14 | OMIM:613291 primary bile acid malabsorption |
| rs55971546 | chr13:103065958 | T/C | 4.1 | Val98Leu | SLC10A2 | 1 (0/1/0/0) | 1.15 [1.10,1.20] | $1.8 \times 10^{-10}$ | 0.97 | OMIM:613291 primary bile acid malabsorption |
| rs2291428 | chr10:45463408 | C/G | 22 | Phe277Leu | MARCH8‡ | 42 (0/2/2/38) | 1.12 [1.10,1.15] | $2.4 \times 10^{-27}$ | 0.55 | Associated with lipid levels |
| rs2290846 | chr4:150277928 | A/G | 28 | Ser2797Leu | LRBA‡ | 25 (0/2/4/19) | 1.12 [1.09,1.14] | $4.7 \times 10^{-27}$ | 0.23 | |
| rs601338*** | chr19:48703417 | G/A | 39 | Trp154Ter | FUT2 | 48 (1/1/43/3) | 0.91 [0.90,0.93] | $9.4 \times 10^{-22}$ | 0.02 | Lewis and ABO(H) histo-blood group antigen secretor allele; cis-eQTL associated with decreased expression of FUT2 in esophagus (mucosa) and other tissues# |
| rs708686** | chr19:5840608 | T/C | 23 | upstream | FUT6 | 1 (0/0/1/0) | 1.26 [1.20,1.32] | $7.4 \times 10^{-20}$ | 0.84 | |
| rs34851490 | chr19:45881296 | G/A | 8.5 | downstream | IRF2BP1 | 28 (0/0/17/11) | 1.12 [1.09,1.16] | $4.2 \times 10^{-15}$ | 0.82 | |
| rs1169288 | chr12:120978847 | C/A | 31 | Ile27Leu | HNF1A | 21 (0/1/2/18) | 0.92 [0.91,0.94] | $1.6 \times 10^{-14}$ | 0.07 | Master regulator of liver specific genes |
| rs13280055 | chr8:11664844 | A/G | 13 | intergenic | - | 3 (0/0/0/3) | 1.11 [1.08,1.14] | $6.9 \times 10^{-14}$ | 0.90 | |
| rs174567 | chr11:61825533 | A/G | 39 | upstream | FADS2 | 59 (0/0/29/30) | 1.07 [1.05,1.09] | $2.3 \times 10^{-12}$ | 0.89 | Bile acid metabolism |
| rs11012737 | chr10:21560840 | A/G | 27 | downstream | MLLT10 | 89 (0/0/14/75) | 1.07 [1.05,1.10] | $3.7 \times 10^{-12}$ | 0.37 | |
| rs2469991 | chr8:119335236 | T/A | 32 | intergenic | MAL2 | 31 (0/0/3/28) | 0.93 [0.91,0.95] | $9.2 \times 10^{-12}$ | 0.60 | Cis-eQTL associated with decreased expression of MAL2 in adipose tissue# |
| rs1935 | chr10:63168063 | C/G | 48 | Glu2353Asp | JMJD1C | 166 (0/1/11/154) | 1.07 [1.05,1.09] | $9.2 \times 10^{-12}$ | 0.04 | Associated with lipid levels |
| rs17240268 | chr15:89804583 | A/G | 12 | Ala311Val | ANPEP | 16 (0/1/3/12) | 0.90 [0.87,0.93] | $6.0 \times 10^{-11}$ | 0.12 | Promotes in-vitro cholesterol crystallization (PMID:8102610) |
| rs12004 | chr22:38481456 | G/T | 31 | Val199Gly | KDELR3 | 237 (0/1/91/145) | 1.07 [1.04,1.09] | $1.2 \times 10^{-10}$ | 0.78 | |
| rs11641445 | chr16:11545628 | T/C | 34 | downstream | LITAF | 48 (0/0/9/39) | 1.06 [1.04,1.09] | $4.2 \times 10^{-10}$ | 0.57 | Cis-eQTL associated with decreased expression of LITAF in adipose and liver tissue# |
| rs17138478 | chr17:37713312 | A/C | 13 | intron | HNF1B | 1 (0/0/0/1) | 1.09 [1.06,1.12] | $5.1 \times 10^{-10}$ | 0.17 | Master regulator of liver specific genes |
| rs2292553 | chr2:218282080 | G/A | 46 | Pro21Leu | TMBIM1† | 166 (0/3/40/123) | 0.95 [0.93,0.97] | $1.1 \times 10^{-8}$ | 0.79 | |
| rs12968116 | chr18:57655270 | T/C | 16 | Arg952Gln | ATP8B1 | 18 (0/1/4/13) | 0.92 [0.90,0.95] | $1.2 \times 10^{-8}$ | 0.26 | OMIM:211600 cholestasis, progressive familial intrahepatic 1 |

*Note*: Samples from Iceland ($N_{cases} = 8757$ and $N_{controls} = 346,688$), UK Biobank ($N_{cases} = 18,417$ and $N_{controls} = 390,150$), in total $N_{cases} = 27,174$ and $N_{controls} = 736,838$. Effect is shown for the minor allele. Significance levels and effects are shown for the combined analysis. LD-class: total number of variants with $R^2 > 0.8$ in the LD-class (HIGH impact variants include stop-gained, frameshift, splice acceptor or donor; MODerate impact variants include missense, splice-region variants and in-frame indels; LOW impact variants include upstream and downstream variants; and LOWEST impact variants include intron and intergenic variants), MAF: Minor allele frequency in Iceland, min: Minor allele, maj: Major allele, OR: Odds ratio, CI: Confidence interval, $P_{het}$: P-value for test of heterogeneity between Iceland and UK. The four low-frequency variants (with MAF < 5%) are shown at the top of the table
*Corresponds to the PI Z allele of *SERPINA1*, **results for *FUT6* are based on a recessive model of inheritance, ***the tested allele rs601338[G] corresponds to the classical wild-type secretor allele of *FUT2*, †the LD-class contains moderate impact variants in two separate genes (a missense variant in *TMBIM1* and two splice-region variants in *ARPC2*), ‡the LD-class contains two moderate impact variants in the same gene, #see Supplementary Table 3 for details on cis-eQTL

and UK). Three novel gallstone-associated non-coding variants at the *LITAF* and *MAL2* loci and a stop gained variant at the *FUT2* locus are correlated with ($r^2 > 0.8$) or represent the strongest cis-eQTLs in their respective regions (Table 1, Supplementary Table 3).

We tested the 32 gallstone disease variants for association with five other related diseases of the biliary system (acute pancreatitis [$N = 3401$], cholecystitis [$N = 3565$], fibrosis/cirrhosis of liver [$N = 1044$], cholestasis of pregnancy [$N = 2615$], and gallbladder cancer [$N = 382$]), six liver biomarkers (alanine aminotransferase [ALT; $N = 172,086$], aspartate aminotransferase [AST; $N = 164,467$], gamma glutamyltransferase [GGT; $N = 156,692$], alkaline phosphatase [ALP; $N = 154,097$], albumin [$N = 92,163$], and bilirubin [$N = 109,748$]), and four lipid traits (high-density lipoprotein [HDL cholesterol; $N = 136,736$], low-density lipoprotein [LDL cholesterol; $N = 126,220$], total cholesterol [$N = 150,211$]) and triglycerides [$N = 119,624$]), resulting in a total of $32 \times (5 + 6 + 4) = 480$ tests that had to be accounted for in the assessment of significance (Supplementary Data 2 and Supplementary Data 3). We observed five significant associations: the Pro290Ser missense variant in *SLC10A2* associates with an increased risk of acute pancreatitis (see below), the previously reported[8] Asp19His missense variant in *ABCG8* also associates with increased risk of acute pancreatitis and cholecystitis, and the two rare coding variants we reported in *ABCB4* (Leu445GlyfsTer22/rs756082276[CCT] andGly622Glu/rs756935975[T]) both associate with high risk of cholestasis of pregnancy, as previously reported[7] (Supplementary Data 2). Ten gallstone disease variants associate with ALT, seven with AST, fourteen with GGT, nine with ALP, two with albumin, and two with bilirubin (Supplementary Data 3). Four variants associate with HDL cholesterol, six with LDL cholesterol, nine with total cholesterol and six with triglycerides (Supplementary Data 3).

The sibling recurrence risk of gallstone disease in the Icelandic population is 1.81 [1.71, 1.91], in line with reported estimates[14] (Supplementary Data 4). Combined, the 32 gallstone disease variants account for a sibling recurrence risk of 1.13 (Supplementary Data 4).

**Low-frequency coding variants in *SLC10A2*.** We found two distinct missense variants in *SLC10A2* (Pro290Ser and Val98Ile; $r^2 < 0.001$) associating with increased risk of gallstone disease (Table 1, Supplementary Table 4). The *SLC10A2* gene encodes the apical sodium-dependent bile acid transporter (ASBT), also known as the ileal sodium-dependent bile acid transporter (ISBT) that mediates the intestinal reabsorption of bile acids.

The variant in *SLC10A2* with the strongest effect on gallstone disease risk is Pro290Ser (rs56398830[A]; OR = 1.36 [1.25–1.49]; $P = 2.1 \times 10^{-12}$, MAF$_{Iceland}$ = 0.6%, MAF$_{UK}$ = 1.1%) (Table 1, Supplementary Fig. 1 and Supplementary Data 1). Others have experimentally shown that Pro290Ser nearly abolishes the bile acid transport activity of ASBT[15,16] (< 3% of the wild-type level[16]). Pro290Ser is in transmembrane domain seven which is highly conserved within the SLC10 family of proteins, and results in a substitution of a non-polar proline with a polar serine[17] (PolyPhen-2: probably damaging, SIFT: deleterious) (Fig. 3a). We observe 62 homozygous carriers of Pro290Ser (expected number of homozygous carriers under Hardy-Weinberg (HW) equilibrium = 55); of whom eight carry a diagnosis of gallstone disease. The gallstone disease odds ratio for homozygous carriers of Pro290Ser is 3.26 [1.55–6.86], consistent with the additive model (Supplementary Fig 2a). Homozygotes are expected to have very little ASBT/SLC10A2 transport activity.

In a meta-analysis of acute pancreatitis in Iceland (1366 cases and 361,081 controls) and UK (1477 cases and 407,090 controls), we detected an association of Pro290Ser with increased risk of this disease (OR = 1.76 [1.38–2.24], $P = 5.6 \times 10^{-6}$, Supplementary Fig. 3 and Supplementary Data 2). Among the 32 gallstone-associated variants, effects on gallstone disease and acute pancreatitis were positively correlated ($r = 0.81$ [0.69–0.92]; $P$

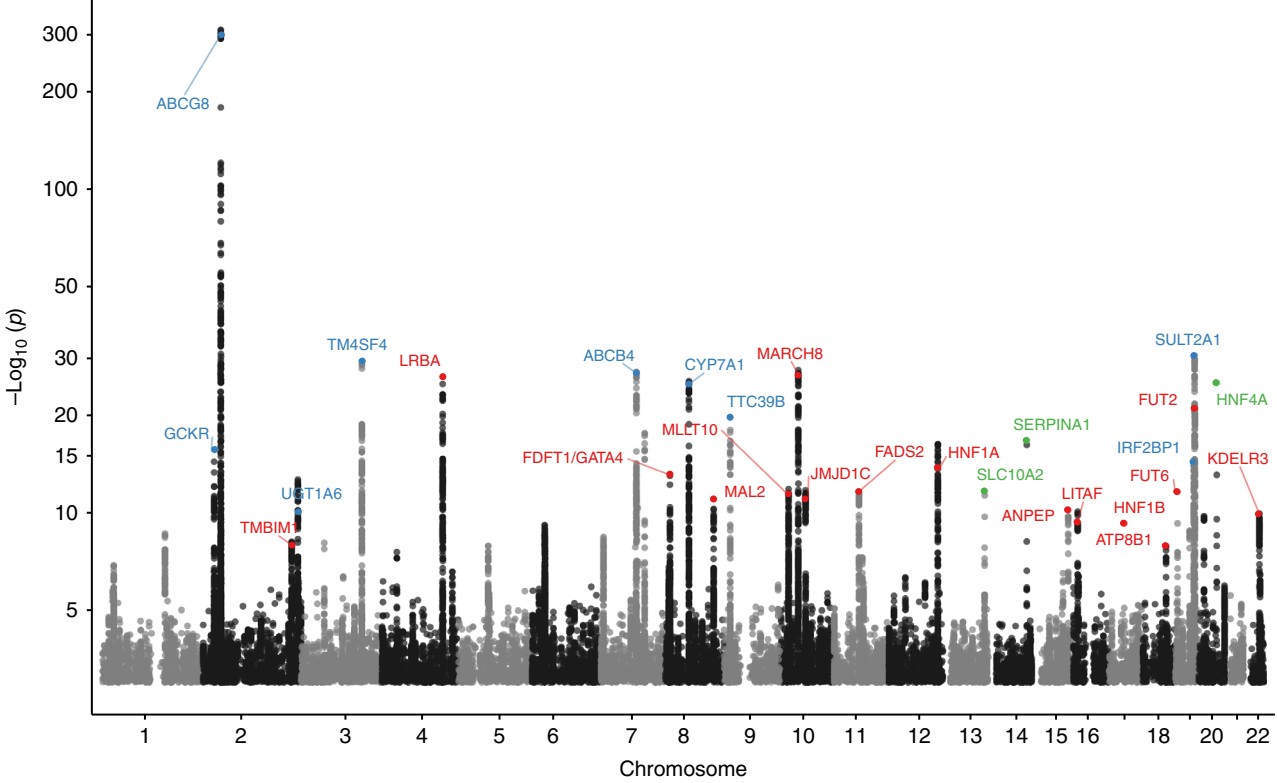

**Fig. 2** Manhattan plot for gallstone disease meta-analysis association results ($N_{cases} = 27,174$ and $N_{controls} = 736,838$). Variants are plotted by chromosomal position (x-axis) and $-\log_{10}$ P-values (y-axis). A chi-square test was used when testing for association. Green = Novel loci represented by low-frequency variants, red = novel loci represented by common variants, blue = reported gallstone disease loci

$= 1.7 \times 10^{-8}$, Supplementary Fig. 3). Since ASBT is a bile acid transporter that mainly transports taurocholate, we tested the effect of Pro290Ser in serum levels of taurocholate ($N = 273$ including four carriers of Pro290Ser). As expected, Pro290Ser associates with reduced levels of serum taurocholate (Effect: $-1.6$ SD $[-2.56, -0.64]$; $P = 0.0010$; Supplementary Table 5). We also tested for an effect of Pro290Ser on fibroblast growth factor 19 (FGF19, $N = 293$), a hormone that regulates bile acid synthesis[18], and observed no effect (Effect: $-0.08$ SD $[-1.02, 0.87]$; $P = 0.87$; Supplementary Table 5). Despite a large sample size, we observed no association with other liver biomarkers in the Icelandic data (Supplementary Data 3).

A more common *SLC10A2* missense variant, Val98Ile (rs55971546[T]), also associates with increased risk of gallstone disease (OR = 1.15 [1.10–1.20], $P = 1.8 \times 10^{-10}$, MAF = 4.1% in both Iceland and the UK) (Table 1, Supplementary Table 4 and Supplementary Data 1). Val98Ile is located in transmembrane domain two and has been shown experimentally by others to reduce the transport activity by 42%[16].

The abovementioned study[16] reported transport activity for Pro290Ser and Val98Ile, as well as two other ASBT/SLC10A2 variants, Val159Ile (rs60380208[T]) and Ser171Ala (rs188096 [A]). For each of these missense variants, we compared the previously reported transport activity measurements to their effect on gallstone disease risk from the current study (Fig. 3b). Lower transport activity is correlated with greater risk of gallstone disease ($r = -0.99$ [$-1.00$, $-0.66$], $p = 8.2 \times 10^{-3}$ for OR vs. activity; $r = -0.99$ [$-1.00$, $-0.77$], $p = 5.2 \times 10^{-3}$ for log(OR) vs. activity). Additionally, we show transport activity compared to the genotypic risk of gallstone disease for different genotypes of Pro290Ser, Val98Ile and wild-type (Supplementary Fig. 4). Together these data indicate that the variants in *SLC10A2* affect

gallstone risk through impairment of the bile acid reabsorption function of ASBT.

We found 17 other missense variants in *SLC10A2* (Supplementary Table 6). Of these, one additional variant, Pro65Leu (MAF$_{Iceland}$ = 0.26%; MAF$_{UK}$ = 0.14%), associates with gallstone disease given the number of tested markers (significance threshold: $P < 0.05/17 = 0.0029$) (OR = 1.40 [1.15–1.72], $P = 0.00096$).

Finally, it is worth noting that a truncating variant in *SLC10A2* (not found in our data) has been claimed to cause hypertriglyceridemia[19]. In our data the SLC10A2 variants Pro290Ser and Val98Ile (which were shown experimentally[16] to have abolished and reduced function, respectively) do not associate ($P > 0.4$ for both variants) with hypertriglyceridemia, defined as having at least two serum triglyceride measurements higher than 5 mmol/L ($N = 1,599$; Supplementary Data 5). Thus, our data do not support the claim that loss-of-function variants in *SLC10A2* cause hypertriglyceridemia.

**Low-frequency coding variant in *SERPINA1*.** A low-frequency missense variant Glu366Lys (rs28929474[T]) in *SERPINA1*, encoding the protease inhibitor alpha-1-antitrypsin (AAT), associates with increased risk of gallstone disease (OR = 1.33 [1.25–1.42], $P = 1.8 \times 10^{-17}$, MAF$_{Iceland}$ = 0.8%, MAF$_{UK}$ = 1.9%) (Table 1, Supplementary Fig. 6 and Supplementary Data 1). Gly366Lys is also known as the Protease Inhibitor (PI) Z allele and causes severe alpha-1 antitrypsin deficiency (AATD) in homozygotes[20]. The most common clinical manifestations of AATD are emphysema and chronic obstructive pulmonary disease (COPD), and a subset of AATD cases develop liver cirrhosis and fibrosis[21]. Consistent with this, in a meta-analysis of Icelandic and UK Biobank emphysema and COPD data, the PI Z allele greatly increases the risk of emphysema (OR = 27.7

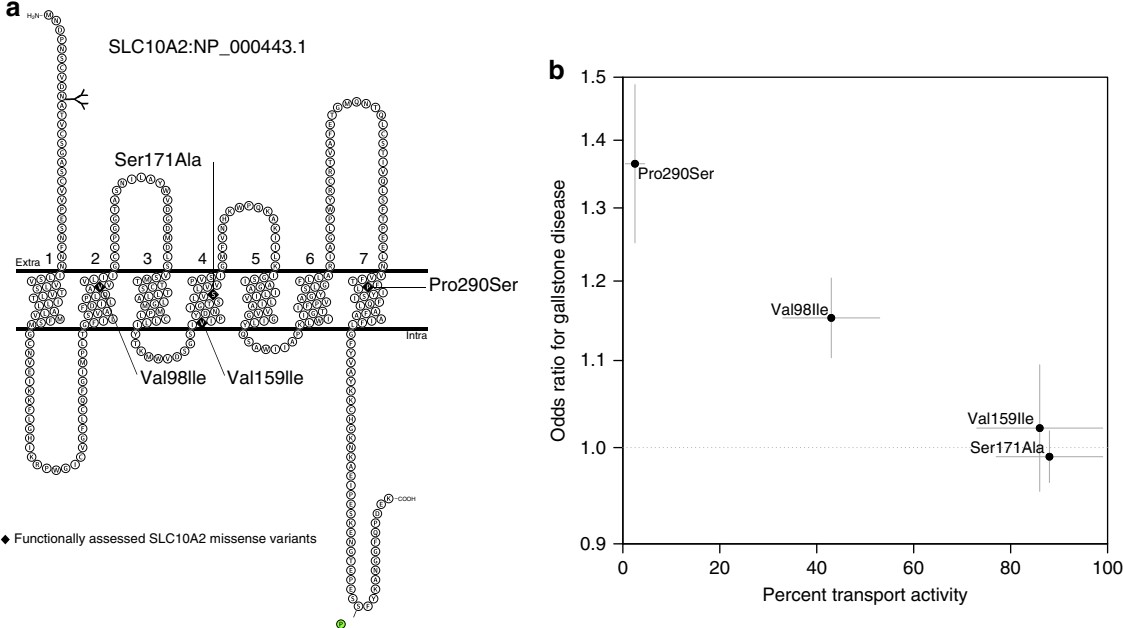

**Fig. 3** SLC10A2 transport activity and gallstone risk. **a** Schematic illustration of the topology of the SLC10A2 protein showing the localization of the functionally assessed missense variants Pro290Ser, Val98Ile, Val159Ile, and Ser171Ala (from PMID: 19823678). **b** A scatter plot showing four SLC10A2 missense variants Pro290Ser, Val98Ile, Val159Ile, and Ser171Ala. The x-axis shows the bile acid transport activity (from PMID:21649730) while the y-axis shows the odds ratio for gallstone disease in the Iceland + UK meta-analysis ($N_{cases} = 27,174$, $N_{controls} = 736,838$)

[17.5–43.7], $P = 6.8 \times 10^{-46}$) and COPD (OR = 4.84 [2.97–7.90], $P = 2.8 \times 10^{-10}$) in homozygotes. As previously reported[22,23], heterozygous carriers of PI Z also are at increased risk of emphysema (OR = 1.44 [1.21–1.70], $P = 3.0 \times 10^{-5}$, Table 2).

We tested the association of PI Z with liver biomarkers in the Icelandic data (Supplementary Table 7 and Supplementary Data 3). PI Z associates with decreased alpha-1-antitrypsin and alpha-fetoprotein levels; and increased levels of alkaline phosphatase, alanine transaminase, gamma glutamyl transpeptidase and increased platelet count, consistent with previous reports[20,24].

We observe four homozygous carriers (PI ZZ) in Iceland and 156 in the UK, compared to 10 and 145 expected under HW equilibrium, respectively. None of the Icelanders and nine of the UK individuals have a gallstone disease diagnosis, and all but one have undergone cholecystectomy indicating symptomatic disease (Supplementary Table 8). To investigate to what extent the observed associations are driven by PI Z allele homozygotes, we calculated the genotypic odds ratios for gallstone disease, emphysema and COPD, based on genotypes of the classical M, S, and Z alleles (Table 2, Supplementary Fig. 2b, 7 and 8). In line with the literature, the genotypic effect was consistent with a recessive trend and a compound heterozygote effect of MZ for emphysema and COPD (OMIM: 613490). However, the genotypic effect on gallstone disease was consistent with an additive model.

In addition to PI Z, the PI S allele in *SERPINA1* (Glu288Val), which is one of the most common AATD deficiency alleles, associates with an increased risk of gallstone disease under the additive model (OR = 1.08 [1.03–1.13], $P = 4.8 \times 10^{-4}$).

**Low-frequency coding variant in *HNF4A***. A low-frequency missense variant, Thr139Ile (rs1800961[T]), in *HNF4A* associates with increased risk of gallstone disease (OR = 1.29 [1.23–1.35], $P = 5.7 \times 10^{-26}$). Thr139Ile has a MAF of 4.6% in Iceland and 3.0% in the UK (Table 1, Supplementary Data 1 and Supplementary Fig. 9). *HNF4A* encodes the hepatocyte nuclear factor 4α, a master regulator of liver-specific gene expression, especially of genes involved in lipid transport and glucose and bile metabolism[25]. Thr139Ile has been associated with a reduction in high density lipoprotein (HDL) cholesterol and total cholesterol levels[26]. In addition, we observe an association of Thr139Ile with increased hemoglobin, bilirubin and gamma glutamyl transpeptidase levels (Supplementary Table 9). In-vitro reporter assays indicate that Thr139Ile decreases the ability of *HNF4A* to activate transcription[27,28].

**Common variant associations**. Seventeen novel common sequence variants (MAF > 5%) associate with gallstone disease (Supplementary Data 1). One of these, rs708686[T] upstream of *FUT6*, fits better a recessive rather than an additive model ($MAF_{Iceland} = 23\%$, homozygous frequency = 5.3%), and associates with increased risk of gallstone disease ($OR_{recessive} = 1.26$ [1.20–1.32], $P_{recessive} = 7.4 \times 10^{-20}$; $OR_{additive} = 1.08$ [1.06–1.10], $P_{additive} = 2.3 \times 10^{-12}$). There is no difference between the risk to heterozygous carriers of rs708686[T] and non-carriers (Supplementary Fig. 10). We have previously reported rs708686[T] to associate with increased levels of vitamin B12[29] (Supplementary Data 6). Similarly with what we see for gallstone disease, we observed a trend towards recessive association of rs708686 with vitamin B12 levels (Supplementary Fig. 11).

Ten of the 17 common signals represent coding variants: one stop-gained (*FUT2*) and nine missense (Table 1 and Supplementary Data 1). We observe an association of the variant rs601338 [G] in *FUT2*, corresponding to the primary Lewis and ABO(H) histo-blood group antigen secretor allele, with reduced risk of gallstone disease (OR = 0.91 [0.90–0.93], $P = 9.4 \times 10^{-22}$). The wild-type secretor allele rs601338[G] has been implicated in susceptibility to gastrointestinal infections[30,31], decreased vitamin B12 levels[32] and with effects on several diseases and other traits[33] (Supplementary Data 6). Consistent with previous observations, rs601338[G] associates with increased risk of gastrointestinal infections (ICD-10 code A08*) in our data (OR = 1.56

**Table 2 Effect of alpha-1-antitrypsin genotypes on A1AT levels, gallstone disease, emphysema and COPD**

| Geno-type | N | Freq. (%) | A1AT levels (median [95% range]) | | Gallstone disease (Iceland + UK) Cases = 27,174, Ctrl = 736,838 | | Emphysema (Iceland + UK) Cases = 3252, Ctrl = 596,760 | | COPD (Iceland + UK) Cases = 15,759, Ctrl = 590,678 | |
|---|---|---|---|---|---|---|---|---|---|---|
| | | | Iceland (N = 5209) | Donato et al.* (N = 21,406) | OR [95% CI] | P | OR [95% CI] | P | OR [95% CI] | P |
| MM | 628,272 | 89 | 157 [105–309] | 149 [100–273] | – | – | – | – | – | – |
| MS | 55,893 | 7.9 | 130 [88–256] | 126 [84–225] | 1.06 [0.99–1.14] | 0.10 | 1.10 [0.97–1.24] | 0.12 | 0.93 [0.88–0.99] | 0.014 |
| MZ | 20,136 | 2.8 | 95 [64–187] | 89 [61–156] | 1.34 [1.26–1.43] | $3.9 \times 10^{-21}$ | 1.44 [1.21–1.70] | $3.0 \times 10^{-5}$ | 0.97 [0.89–1.07] | 0.58 |
| SS | 1248 | 0.18 | 110 [88–147] | 95 [49–181] | 1.11 [0.85–1.45] | 0.44 | 0.83 [0.34–2.00] | 0.68 | 1.10 [0.78–1.55] | 0.60 |
| SZ | 975 | 0.14 | 63 [59–82] | 64 [42–108] | 1.36 [1.04–1.79] | 0.027 | 1.06 [0.44–2.57] | 0.89 | 1.02 [0.68–1.53] | 0.94 |
| ZZ | 164 | 0.023 | –** | 25 [15–57] | 1.29 [0.66–2.52] | 0.46 | 27.7 [17.5–43.7] | $6.8 \times 10^{-46}$ | 4.84 [2.97–7.90] | $2.8 \times 10^{-10}$ |

N number of individuals with each genotype, Freq genotype frequency for combined set, OR odds ratio comparing each genotype to MM, CI 95% confidence interval for the odds ratio
*Donato et al. refer to A1AT measurements in Table 3 of PMID:22912357
**No PI ZZ carriers had A1AT measurements in the Icelandic dataset

[1.34–1.82], $P = 7.2 \times 10^{-9}$) (Supplementary Table 10). In Iceland we also observe an effect of rs601338[G] on decreased gamma glutamyl transferase and increased folate (Supplementary Table 10). *FUT2* is a paralog of the previously mentioned *FUT6* and they both belong to the gene family of fucosyltransferases (63% identical on the protein level).

Of note, Ala311Val (rs17240268[A]) in *ANPEP* (aka *CD13*) associates with decreased risk of gallstone disease (OR = 0.90 [0.87–0.93], $P = 6.0 \times 10^{-11}$). This gene encodes Aminopeptidase N, which has been reported to promote cholesterol crystallization in in-vitro systems[34] and through that may affect cholesterol gallstone formation[35]. The minor allele rs17240268[A] would be expected to have a reducing effect on protein function for the observed protective effect on gallstone disease to be consistent with the reported promoting effect of *ANPEP* on cholesterol crystallization.

In addition to the hepatocyte nuclear factor *HNF4A* (see above), we observe signals in two other hepatocyte nuclear factor genes. The missense variant Ile27Leu (rs1169288[C]) in *HNF1A* associates with decreased risk of gallstone disease (OR = 0.92 [0.91–0.94], $P = 6.0 \times 10^{-11}$). *HNF1A* encodes the hepatocyte nuclear factor 1α and is involved in the regulation of the expression of several liver-specific genes[36] and bile acid transporters, including ASBT, in the enterocytes of the terminal ileum[37]. Ile27Leu has been associated with higher LDL cholesterol levels[38,39] and increased risk of coronary artery disease[40] (Supplementary Data 6). Diabetes, hypercholesterolemia and increased bile acid and cholesterol synthesis were observed in mice deficient for *HNF1A*[37]. An intron variant rs17138478[A] in *HNF1B* associates with increased risk of gallstone disease (OR = 1.09 [1.06–1.12], $P = 5.1 \times 10^{-10}$). The variant falls within a liver specific enhancer element controlling the expression of itself and has *HNF4A* among its DNA binding proteins (Supplementary Data 7).

**Lipid variants and gallstone disease.** The biological functions of the 32 gallstone associated variants were assessed by gene enrichment analysis using ToppGene[41]. Variants were assigned to genes if they were located inside the gene or located within 5 kbs upstream and downstream from a gene, on the basis of mappings from previous GWAS studies or based on evidence from eQTL data (GTEx). In total 31 variants could be assigned to 27 genes that were included in the subsequent analysis (see Supplementary Data 8 for inclusion criteria). The analysis revealed enrichment of genes at loci reported to associate with lipid levels (PMID:24097068, $P = 2.5 \times 10^{-18}$, 11 out of 212 genes), in addition to genes linked to bile secretion (KEGG:193146, $P = 7.6 \times 10^{-8}$, 5 out of 71 genes) and maturity onset diabetes of the young (KEGG:83096, $P = 9.0 \times 10^{-6}$, 3 out of 26 genes). Based on these

results and because of the high cholesterol content of most gallstones, we tested 203 variants representing signals reported to associate with either HDL cholesterol, LDL cholesterol, total cholesterol or triglyceride serum levels for risk of gallstone disease (Supplementary Data 9). There is an excess of gallstone associated variants among reported HDL cholesterol, LDL cholesterol and total cholesterol associated variants: eight out of 83 HDL cholesterol variants ($P = 5.3 \times 10^{-19}$ based on a binomial test, see Methods), nine out of 65 LDL cholesterol variants ($P = 1.1 \times 10^{-22}$), fourteen out of 77 total cholesterol variants ($P = 2.5 \times 10^{-36}$) and five out of 49 triglyceride variants ($P = 1.2 \times 10^{-12}$) also associate with gallstone disease when all 203 tests are accounted for in the assessment of significance. In total, 23 of the 203 unique variants associating with one or more of HDL cholesterol, LDL cholesterol, total cholesterol and triglycerides also associate with gallstone disease. However, there is no consistency in the directions of the effects on serum cholesterol and gallstone disease risk (Fig. 4). This indicates that lipid serum levels are not by themselves causative factors in gallstone formation, even though cholesterol metabolism appears to have an impact on gallstone risk.

Of note is that a missense variant Gln535Arg in *DAGLB* (rs2303361[C]), encoding diacylglycerol lipase beta, associates with an increased risk of gallstone disease (OR = 1.06 [1.04–1.08], $P = 4.7 \times 10^{-8}$). The variant is in strong LD ($r^2 = 0.95$) with the intron variant rs1880118[C] that was recently reported to associate with increased HDL cholesterol levels and increased *DAGLB* expression in subcutaneous adipose tissue[42]. We replicate the association of rs1880118[C] with increased HDL cholesterol in our data (effect = 0.15 SD [0.10–0.20], $P = 2.4 \times 10^{-9}$).

**Diabetes and gallstone disease.** The relationship between gallstone disease and diabetes is not clear[43]. To explore whether there is a difference in the effect of gallstone associated sequence variants between diabetics and non-diabetics, we tested whether the 32 gallstone disease associated variants confer different risk on diabetics and non-diabetics and found no significant differences (Supplementary Fig. 12). To further explore the relationship of gallstone disease and genes linked to maturity onset diabetes of the young (MODY), we tested 12,809 variants in 12 genes linked to MODY (including 36 known pathogenic MODY variants) for association with gallstone disease; no additional signals associating with gallstone disease were observed (Supplementary Data 10).

**Self-report vs. hospital diagnosis.** Since 4067 of the 18,417 UK cases are based only on self-reporting, we have looked for a difference between association results for self-reported and ICD10-diagnosed cases. There is no difference between effect sizes in the

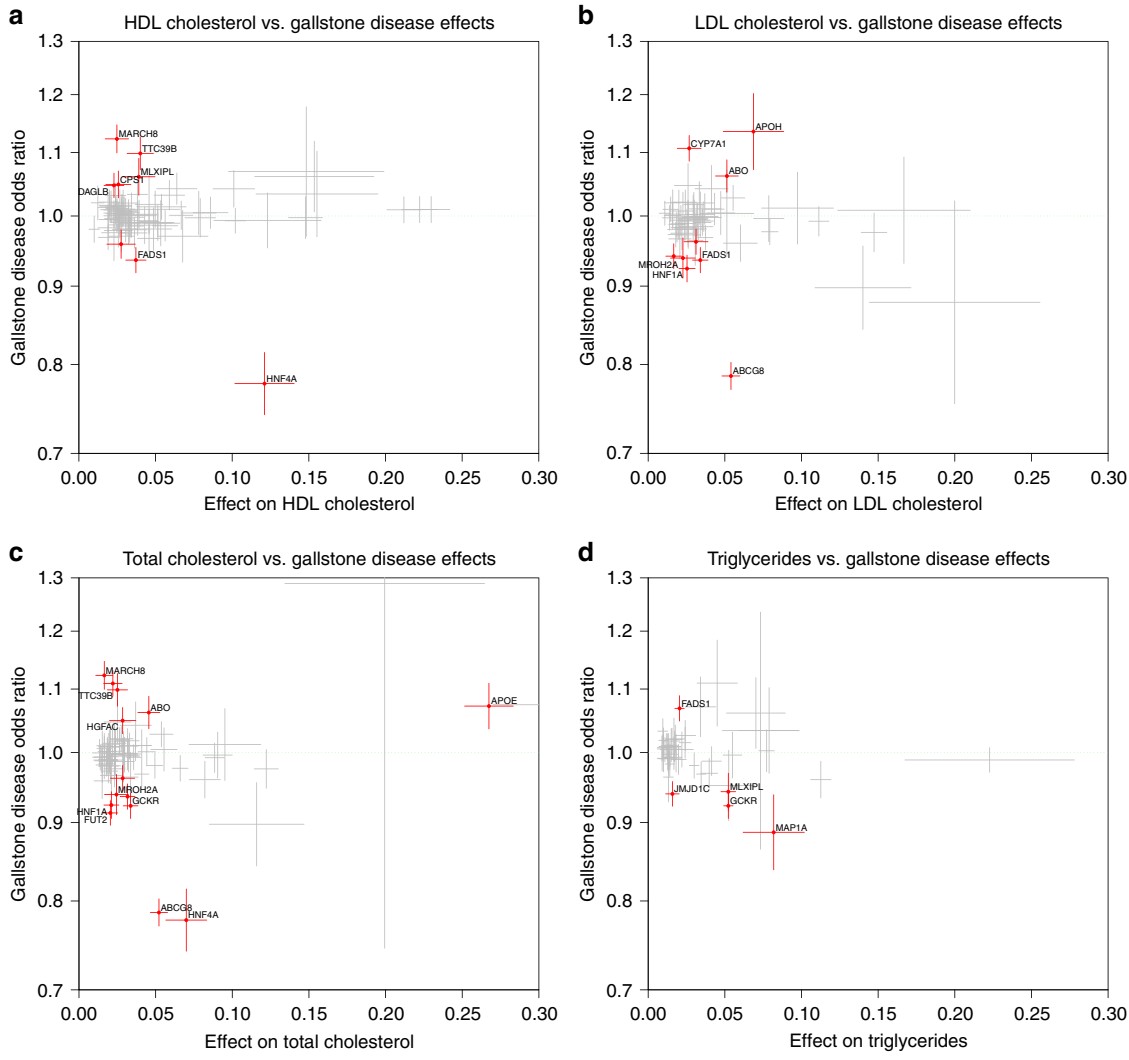

**Fig. 4** 203 variants reported to be associated with either HDL cholesterol, LDL cholesterol, total cholesterol or triglyceride levels: Effect of the allele associated with an increased level of serum lipids vs. odds ratio (OR) for gallstone disease, for **a**) HDL cholesterol, **b**) LDL cholesterol, **c**) total cholesterol, and **d**) triglycerides. Crosses indicate 95% confidence intervals. Variants with a significant OR for gallstone disease ($P < 0.05/203 = 2.6 \times 10^{-4}$) are shown in red

two groups (ICD10 and self-reported) for 30 of the 32 variants found. The two variants with a significant difference between ICD10-based and self-reported-based effects are the novel gallstone disease associated variant *SERPINA1* Glu366Lys (PI Z; rs28929474[T]) (*P*-value for test for heterogeneity: $P_{het} = 2.2 \times 10^{-4}$) and the previously reported *ABCG8* Asp19His (rs11887534 [C], $P_{het} = 1.6 \times 10^{-9}$) missense variants, which both showed a larger effect when using self-reported cases than ICD10 code cases. However, both these variants associate with both ICD10-based and self-reported gallstone disease (Supplementary Fig. 13, Supplementary Table 11).

## Discussion
Through a GWAS of gallstone disease in Iceland and UK, we found 21 novel sequence variants that associate with the disease, including four low frequency coding variants in *SLC10A2* (two distinct variants), *HNF4A* and *SERPINA1* with relatively large effects.

Most notably, we demonstrate that two distinct low-frequency missense variants in the bile acid transporter *SLC10A2* — Pro290Ser and Val98Ile — associate with higher

risk of gallstone disease. The variants have experimentally been shown to reduce the transport activity of the protein encoded by the gene, commonly known as ASBT[16]. We observe a tendency to a higher risk of gallstone to associate with a lower transport activity of different variants of *SLC10A2*. The main function of ASBT is to reabsorb bile salts (up to 95%) from the terminal ileum into ileocytes, after which the bile salts are transported back to the liver through the enterohepatic circulation[44]. We postulate that impairment of ASBT leads to disturbances of the enterohepatic circulation, causing imbalances of the relative amounts of bile acids, cholesterol and phospholipids, thus increasing the risk of gallstone formation. Because of its role in the enterohepatic circulation, a connection between *SLC10A2* and gallstone disease risk has been postulated. In a candidate gene study[45] with 240 gallstone cases and 255 controls, the rs9514089[C] *SLC10A2* intron variant was reported to associate with an increased risk of gallstone disease (OR = 2.04 [1.19–3.55]; $P = 7.7 \times 10^{-3}$). The reported p-value does not survive multiple testing correction accounting for the 30 markers tested in their manuscript. A subsequent study[46] found no association of rs9514089 with gallstone disease (OR

= 0.83 [0.63–1.09]; $P = 0.19$), and this variant has not been reported to associate with gallstone disease by others[7–10]. In our Icelandic and UK data (with 27,174 cases) we find no association between rs9514089[C] and gallstone disease risk (OR = 0.991 [0.973–1.010]; $P = 0.36$; Supplementary Table 12). Thus, we conclude that the claimed association of rs9514089 with gallstone disease is a false positive. Other coding variants in *SLC10A2* have been implicated in primary bile acid malabsorption, based on a single pedigree with one identified compound heterozygous carrier of a missense and a splice-donor variant[47]. *SLC10A2* inhibitors have previously been considered as a therapeutic candidate for type 2 diabetes[48], but are now pursued as a treatment of chronic constipation due to the initially unwanted diarrhea and steatorrhea side-effects[49]. Our results indicate that inhibition of *SLC10A2* associates with higher risk of gallstone disease as homozygotes for Pro290Ser (expected to have less than 3% transport activity[16]) have an odds ratio of 3.26 [1.55–6.86]. The genes *ABCB4* and *SULT2A1*, previously reported to be associated with gallstone disease, are also involved in bile homeostasis[8,13].

A number of the novel variants described here to associate with gallstone disease are at loci linked to Mendelian diseases: Interestingly, the hepatocyte nuclear factors *HNF1A*, *HNF1B,* and *HNF4A* are all linked to autosomal dominant forms of maturity-onset diabetes of the young (MODY3, OMIM:600496; MODY5, OMIM:137920; and MODY1, OMIM:125850; respectively). In addition, among the reported gallstone loci, the glucokinase regulatory protein (*GCKR*) is a regulator of glucokinase (*GCK*), that is linked to MODY2 (OMIM:125851).

Among the novel gallstone variants, two are in fucosyltransferase genes (*FUT2* and *FUT6*). Fucosyltransferases mediate fucosylation which is an abundant posttranslational modification of glycosylated proteins and lipids. *FUT2* is responsible for the majority of fucosylation in the gastrointestinal tract[50,51]. Interestingly, we observe opposing effects of the *FUT2* variant rs601338 on the risk of gallstone disease and susceptibility to gastrointestinal infection. This constitutes an example of antagonistic pleiotropy where a variant has opposing effects on the risk of two diseases. The increased risk of gastrointestinal infection observed for the wild-type secretor allele rs601338[G] is likely due to the presence of fucosylated glycans in the gastrointestinal tract that serve as adhesion sites for pathogens. *FUT2* may influence biliary disease progression through interaction with the biliary and intestinal microbiota[52,53]. However, the mechanism through which the rs601338 variant influences the risk of gallstone disease is unclear, as the *FUT2* locus is highly pleiotropic and is the subject of balancing selection in humans[54,55]. Further investigation is needed to explain the role of fucosylation in the pathogenesis of gallstone disease.

We show that the PI Z allele of *SERPINA1* associates with increased risk of gallstone disease. Severe AAT deficiency in PI ZZ homozygotes predisposes to emphysema and, less commonly, to liver disease[20]. The precise pathomechanism of liver disease associated with severe AAT deficiency is not well understood but histopathological findings of liver parenchyma in PI ZZ homozygotes indicate that liver abnormalities result from toxic intracellular accumulation of AAT in hepatocytes[56]. Moderate AAT deficiency in PI MZ heterozygotes has been associated with slightly increased risk of liver disease and intracellular accumulation of AAT in hepatocytes[57]. We speculate that MZ heterozygosity may result in general liver dysfunction as result of toxic accumulation of AAT in hepatocytes over time, thus predisposing to gallstone disease.

Biliary cholesterol secretion plays a major role in cholesterol homeostasis[58]. Variants in genes involved in cholesterol homeostasis, including *ABCG5/ABCG8* and *CYP7A1*, are reported to

associate with gallstone disease[8]. Six of the novel gallstone associated variants, or highly correlated variants ($r^2 > 0.8$), have been reported to associate with blood cholesterol levels (in *HNF4A*, *HNF1A*, *FUT2*, *FADS2*, *MARCH8*, and *JMJD1C*) (Supplementary Data 9). Most notably, *HNF4A* is a nuclear receptor highly expressed in the liver, and has been shown to control the expression of the closely linked cholesterol transporters *ABCG5* and *ABCG8* that share a promotor[59]. Despite a clear role of the aforementioned genes in cholesterol homeostasis, it remains unclear whether serum cholesterol levels affect risk of gallstone disease[60]. When considering 203 variants reported to associate with serum HDL cholesterol, LDL cholesterol, total cholesterol or triglyceride levels we do not observe a consistent direction of effect of lipid increasing alleles on the risk of gallstone disease. This suggests that there is not a direct path between serum lipid levels and the production of gallstones, which is in line with observations from previous studies[61].

While our data, as well as data from others[61], do not support a direct path between serum cholesterol levels and gallstone formation, some aspects of cholesterol metabolism may contribute to gallstone disease. Biliary cholesterol super-saturation is considered a major factor in promoting cholesterol nucleation and gallstone formation[62–64]. Thus sequence variants affecting the amount of cholesterol secreted into bile, or the cholesterol/bile acid ratio, are likely to impact gallstone formation. Depending on the mechanism of action of the cholesterol regulating genes, the allele associated with gallstone risk may not consistently decrease or increase serum cholesterol levels. For example, increased activity of the sterol-transporter *ABCG5/8* that has a direct role in promoting biliary cholesterol secretion and in reducing intestinal absorption of dietary cholesterol is expected to increase the risk of gallstones while lowering circulating cholesterol. In contrast, reduced function of *CYP7A1*, the rate-limiting enzyme in the conversion of cholesterol to bile acids in the liver, likely predisposes to gallstone formation through a decrease in bile acid synthesis. It has been postulated that the increase in serum cholesterol levels that has been observed in carriers of *CYP7A1* gallstone risk variants may be mediated through down-regulation of hepatic LDL receptors[65].

In summary, we have discovered 21 novel gallstone disease variants in the largest gallstone disease GWAS to date. The associations presented emphasize the role of sequence variants in genes involved in cholesterol homeostasis and specifically highlight the intestinal compartment of the enterohepatic circulation in the pathogenesis of gallstone disease. We conclude that sequence variants affecting the amount of cholesterol secreted into bile, or the cholesterol/bile acid ratio, are likely to cause gallstone formation.

## Methods

**Study subjects from Iceland**. This study is based on whole-genome sequence data from the whole blood of 15,220 Icelanders participating in different disease projects at deCODE genetics. Additionally, 151,677 Icelanders have been genotyped using Illumina SNP chips and genotype probabilities for untyped relatives have been calculated based on Icelandic genealogy. The process used to whole-genome sequence the Icelandic population, and the subsequent imputation from which the data for this analysis were generated, has been comprehensively described in a recent publication[7]. We note that selection of individuals for whole-genome sequencing and chip genotyping was not performed with regards to gallstone case status.

All individuals who donated blood, or their guardians, provided written informed consent. All sample identifiers have been encrypted in accordance with the regulations of the Icelandic Data Protection Authority. The study was approved by the National Bioethics Committee (ref:VSNb2015100030/03.03). Personal identities of the participants and biological samples were encrypted using a third-party system approved and monitored by the Icelandic Data Protection Authority. The National Bioethics Committee approved the study, including the protocol, methodology and all documents presented to the participants. All methods were performed in accordance with the relevant guidelines and regulations.

To identify gallstone cases, we searched for patients with International Classification of Diseases (ICD) codes, diagnosis code ICD-10 K80 Cholelithiasis and ICD-9 574, indicative of gallstone disease, at Landspitali—The National University Hospital of Iceland in Reykjavik (LUH), and the Icelandic Medical Center in Mjodd (Laeknasetrid). A total of 8757 gallstone cases were included in the association analysis; 892 of these were whole-genome sequenced, 5840 were genotyped using various Illumina chips and imputed using long-range phased haplotypes, and genotype probabilities for 2026 were imputed on the basis of information from genotyped close relatives. In addition, cases of five diseases of the biliary system were identified by the ICD codes K85 for acute pancreatitis, K81 for cholecystitis, K74 for fibrosis/cirrhosis of liver, O26.6 for cholestasis of pregnancy and C23 for gallbladder cancer. Individuals recruited through different genetic research projects at deCODE were used as controls, and individuals in the gallstone cohort were excluded from the control group. Among the controls, 13,204 were whole-genome sequenced, 133,026 were genotyped by chip, and 200,458 were imputed on the basis of the genotypes of close relatives. In total, the number of controls was 346,688.

**Study subjects from the UK**. The UK Biobank project is a large prospective cohort study of over 500,000 individuals from across the United Kingdom, aged between 40–69 at recruitment[66]. Genotyping was performed using a custom-made Affimetrix chip, UK BiLEVE Axiom[67] in the first 50,000 participants, and with Affimetrix UK Biobank Axiom array in the remaining participants[68], 95% of the signals are on both chips. Imputation was performed by Wellcome Trust Centre for Human Genetics using the Haplotype Reference Consortium (HRC) and the UK10K haplotype resources[11]. This yields a total of 96 million imputed variants, however only 40 million variants imputed using the HRC reference set were used in this study due to quality issues with the remaining variants.

The population from UK Biobank consisted of 18,417 cases and 390,150 controls, all individuals of European ancestry. Gallstone disease was ascertained based on self-reported (self-reported non-cancer illness, Data-Field 20002) and ICD diagnoses obtained from primary or secondary diagnoses codes a participant has had recorded across all their episodes in hospital. Diagnoses are coded according to ICD-9 code 574 or ICD-10 code K80. The UKB Resource was accessed under Application Number '24711'.

**Association testing and meta-analysis**. We used logistic regression to test for association between variants and disease, assuming a multiplicative model, treating disease status as the response and expected genotype counts as covariates. For the Icelandic cohort, the association testing was done using software developed at deCODE genetics[7]. The threshold for genome-wide significance was corrected for multiple testing using a class-specific Bonferroni procedure based on predicted functional impact of classes of variants[12]. This yielded significance thresholds of $2.0 \times 10^{-7}$ for high-impact variants (including stop-gained, frameshift, splice acceptor or donor, $N = 11,465$), $3.9 \times 10^{-8}$ for moderate-impact variants (including missense, splice-region variants and in-frame indels, $N = 197,583$), $3.6 \times 10^{-9}$ for low-impact variants (including upstream and downstream variants, $N = 2,971,445$) and $5.9 \times 10^{-10}$ for lowest-impact variants (including intron and intergenic variants, $N = 39,726,619$). We have previously demonstrated that the variant annotation based multiple testing method provides greater power to detect associations than Bonferroni correction where all variants are treated equally, while maintaining an overall family-wise error rate of 0.05. All variants except loss-of-function variants (including stop-gained, frameshift, splice acceptor or donor) have $p$-value thresholds stricter than the common threshold of $5.0 \times 10^{-8}$, while loss-of-function variants were given a threshold of $2.0 \times 10^{-7}$ due to their higher prior probability of representing true positive signals. In total, 42,907,111 variants were tested in the meta-analysis. Variants in the UK Biobank imputation dataset have been mapped to NCBI Build38 positions and subsequently matched to the variants in the Icelandic dataset based on allele variation. Results from UK and Iceland were combined using a Mantel-Haenszel model[69] in which the groups were allowed to have different population frequencies for alleles and genotypes but were assumed to have a common OR. We tested for heterogeneity by comparing the null hypothesis of the effect being the same in all populations to the alternative hypothesis of each population having a different effect using a likelihood ratio test. $I^2$ describes the proportion of total variation in study estimates due to heterogeneity.

**Sibling recurrence risk**. Sibling recurrence risk ratios ($\lambda S[i]$) for the risk alleles of the identified signals associated with gallstone disease in the present meta-analysis. We estimated the risk ratio among siblings ($\lambda S$) of Icelandic patients with gallstone disease ($N = 2628$) by cross-matching with a genealogy database that covers the entire Icelandic nation. Risk ratio among siblings ($\lambda S$) was estimated at 1.81 [1.71, 1.91] ($P < 0.0001$), using the approach suggested by Edvardsson et al.[70]. Also, we calculated the proportion of sibling recurrence risk of gallstone disease explained by the signals identified in the present study, which is given by $\log(\lambda S[i])/\log(\lambda S)$[71].

**Population attributable fraction**. Population attributable fraction is defined as the fraction of cases that would be eliminated from the population if the risks of all individuals carrying the risk variant could be contained, e.g., through a treatment, to be the same as non-carriers of the at-risk variant(s). It can be calculated for a variant using the following formula:

$$\text{PAF} = 1 - (1/W) \text{ where } W = (1 - p)^2 + 2p(1 - p)\text{RR} + p^2\text{RR}^2$$

Here 'p' denotes the at-risk allele frequency and RR is the relative risk of for a disease. Relative risks are estimated by odds ratios under the assumption of the multiplicative model. Accordingly, the RR for carrying two risk variants is $\text{RR}^2$.

**Gene set enrichment analysis**. Gene set enrichment analysis was conducted using the online ToppGene tool[41].

**Binomial test for excess of cholesterol variants**. Excess of gallstone associated variants among reported HDL cholesterol, LDL cholesterol and total cholesterol variants was tested using a binomial model, as follows: Since 170 variants were tested, a Bonferroni adjusted significance level of $0.05/170 = 2.9 \times 10^{-4}$ was used. Thus, under the null hypothesis of no association with gallstone disease, the probability of getting a significant association is $2.9 \times 10^{-4}$ for each variant, since the $P$-values for association with gallstone disease have a Uniform[0,1] distribution under the null. Therefore, the number of significant associations with gallstone disease in each group has a Binomial (N, P) distribution, where N is the number of tested variants in the group and $P = 2.9 \times 10^{-4}$, and this can be used as the basis for a binomial test. For example, for the HDL cholesterol variants, eight out of 83 variants associate with gallstone disease, and the $P$-value for the binomial test of excess of HDL variants is therefore $\text{Prob}(X \geq 8) = 2.2 \times 10^{-18}$, where X has a Binomial($83, 2.9 \times 10^{-4}$) distribution.

**URLs**. For Online Mendelian Inheritance in Man (OMIM), see https://www.omim.org/. For GWAS catalog, see https://www.ebi.ac.uk/gwas/. For GTEx Portal, see https://www.gtexportal.org/.

**Code availability**. We used publicly available software (URLs listed below) in conjunction with the above described algorithms in the sequencing processing pipeline (Whole-genome sequencing, Association testing, RNA-seq mapping and analysis): for BWA 0.7.10 mem, https://github.com/lh3/bwa; for GenomeAnalysisTKLite 2.3.9, https://github.com/broadgsa/gatk/; for Picard tools 1.117, https://broadinstitute.github.io/picard/; for SAMtools 1.3, http://samtools.github.io/; for Bedtools v2.25.0-76-g5e7c696z, https://github.com/arq5x/bedtools2/; for Variant Effect Predictor, https://github.com/Ensembl/ensembl-vep; for BOLT-LMM, https://data.broadinstitute.org/alkesgroup/BOLT-LMM/downloads/.

## Data availability

Sequence variants passing GATK filters have been deposited in the European Variation Archive, accession number PRJEB15197.

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

## Acknowledgements
We thank the individuals who participated in this study and whose contribution made this work possible. We also thank our valued colleagues who contributed to the data collection and phenotypic characterization of clinical samples, as well as to the genotyping and analysis of the data.

## Author contributions
E.F., A.O., A.M.D., A.H., H.H., D.F.G., P.S., and K.S. designed the study and interpreted the results. S.G., I.J., S.O., T.S., and E.S.B. carried out subject ascertainment and recruitment. I.O., G.I.E., and O.S. contributed to data acquisition. E.F., A.O., S.B., G.T., A. M.D., S.J., O.A.S., F.Z., B.G., G.H.H., G.S., D.A.S., G.M., D.F.G., and P.S. performed statistical and bioinformatics analyses. E.F., A.O., S.B., G.T, A.M.D., O.A.S., G.L.N., G.A. A., B.O.J., R.P.K, T.R., U.T., D.F.G., P.S., and K.S. drafted the manuscript. All authors contributed to the final version of the paper.

## Additional information

**Competing interests:** The authors E.F., A.O., S.G., S.B., G.T., A.M.D., S.J., O.A.S., G.L.N., F.Z., G.A.A., B.G., G.H.H., A.H., B.O.J., R.P.K., G.S, D.A.S., G.M., H.H., I.J., T.R., U.T., D. F.G., P.S., and K.S., who are affiliated with deCODE genetics/AMGEN, declare competing financial interests as employees. The remaining authors declare no competing financial interests.

