## [Peer Review File · Nature Communications]

Reviewer #1 (Remarks to the Author):

Overall:

Ferkingstad E and colleagues perform a comprehensive genome-wide association study for gallstone disease in deCODE and the UK Biobank, comprising of 27,174 cases and 736,838 controls, and the largest GWAS for this condition to date. Recently, Joshi AD et al Gastroenterology 2016 reported a GWAS for gallstones in 8,720 cases and 55,152 controls (discovery) and 6,489 cases and 62,797 (controls). They replicate these findings as well as from other studies and demonstrate new associations. The study was comprehensively and rigorously performed by an excellent research team.

Beyond performing a larger GWAS, the authors do not convey new insights. The last sentence of Joshi AD et al's abstract reads: "In this large-scale GWAS of gallstone disease, we identified 4 loci in genes that have putative functions in cholesterol metabolism and transport, and sulfonation of bile acids or hydroxysteroids." This paper's abstract's last sentence reads: "Taken together with replication of previously reported variants, this highlights the role of cholesterol metabolism and the enterohepatic circulation of bile acids in gallstone disease." The conclusions essentially are the same and highlights a major deficiency of the paper. While novel loci and variants are discovered, there are no novel insights.

As before, the authors demonstrate an enrichment of cholesterol-related genes. And, as before, the authors demonstrate that serum cholesterol is unlikely to be causal (Stender S et al. J Hepatol. 2013). While the authors now demonstrate the association of low-frequency coding mutations at SLC10A2 are associated with gallstone disease, prior common variants were previously described in smaller studies.

Instead of focusing on the long list of associations and biological plausibility, additional analyses to characterize causal pathways complementary to enrichment analyses would be helpful. For example, what aspects of cholesterol metabolism are relevant, particularly if serum cholesterol is not causal? Can the relationship with diabetes be further explored? What is the difference between gallstone variants that also associate with pancreatitis versus not?

Major:

1. I am a bit confused why SLC10A2 is considered a novel locus for gallstones. Prior human genetic studies have implicated this gene for gallstones in other, albeit much smaller, studies, which the authors appropriately acknowledge. Are the authors distinguishing novelty based on variants or loci or both? The Manhattan plot (Figure 2) is confusing because it uses a mix of this terminology.

2. Hypertriglyceridemia is co-morbid with gallstones as well as pancreatitis. Additionally, disruptive mutations in SLC10A2 have been linked to severe hypertriglyceridemia (Love MW et al. ATVB 2001). Why were triglycerides not included in the lipids analyses?

3. For the gene enrichment analyses, how are genes identified/prioritized for non-coding analyses? Additional (brief) details would be helpful instead of only referring the reader to the ToppGene paper.

4. The observation of showing enrichment with cholesterol-related variants but not having consistency with respect to effect direction for gallstone risk is interesting and has been observed, which the authors acknowledge. It likely suggests that these lipid values are not causal, as the authors mention. But, in the same sentence they also say “even though cholesterol metabolism has an impact on gallstone risk.” It would be helpful to further explore this. A major claim of this paper is demonstrating the importance of cholesterol metabolism. If cholesterol-related lab values do not appear to be causal, are there aspects of cholesterol metabolism that may be important/causal?

5. Figure 3B does not clearly demonstrate a dose dependency in SLC10A2 transport activity with odds for gallstone disease. I would consider an alternative figure or move to the Supplement. The authors also make the claim in the Discussion that there is a dose-dependent relationship. In Figure 3B, I just see substantially elevated risk with nearly 0% enzymatic activity and a non-linear relationship for the rest of the variants. Is gross SLC10A2 activity relevant or are disrupting particular domains more relevant?

6. Given the epidemiological relationship with diabetes as well as implication of MODY genes, are there variants that have differential risk on gallstone disease among diabetics vs non-diabetics?

Minor:

1. In the Abstract’s second sentence, it is stated: “Due to the threat of complications, such as acute pancreatitis, it is commonly treated with cholecystectomy.” This is not true for asymptomatic, or incidental, gallstones.

2. SERPINA1 and HNF4A should be italicized in the Abstract. Please check the manuscript throughout to ensure all gene names are italicized.

3. Please clarify whether genotyping or sequencing strategy was performed with regards to gallstone case status.
4. Why is there a roughly 2-fold prevalence of gallstones in the UK Biobank vs deCODE? Perhaps a Supplementary Table with demographics and relevant comorbidities could be helpful.
5. Suggest changing “HDL” to “HDL cholesterol” and “LDL” to “LDL cholesterol.”
6. On page 4, what is meant by “unrelated missense variants?”

Reviewer #2 (Remarks to the Author):

This is an interesting study by Ferkingstad et al. performing a genome-wide association meta-analysis for gallstone disease utilizing large cohorts from Iceland and UK (total n=27,174 cases and 736,838 controls). The study identified 21 novel variants at 20 loci for gallstone disease. While these results are of interest in further elucidating the genetic component of gallstone disease, I have some comments that would need clarification, especially regarding the diagnostic criteria, novelty of the study, and the details how multiple testing was corrected.

1. It appears that the controls were not screened for gallstone disease. The authors should give a justification for this design and its potential effects on the results.

2. How many of the 18,417 UK cases were diagnosed based on only self-reporting? To assess the robustness of self-reported diagnosis of gallstone disease, it would be important to check if the GWAS association results for gallstone disease were similar in the UK participants when using only the self-reported cases versus using the cases with the ICD diagnoses made by the physicians at the hospitals?

3. It is not clear why the authors selected the variant annotation based design to correct for multiple testing (ref 12). The rationale for this should be provided.

4. The authors describe results from a recessive model for SNP rs708686, upstream of FUT6. How many statistical models were tested all in all? Were the presented results also corrected for the number of these tested models?

5. Previous studies have shown that variant Pro290Ser abolishes the bile acid transport activity of ASBT. It would be important for the reader that the authors would explain throughout the manuscript which results are new and which support previous data. Functional studies with the novel variants would expand the results beyond a GWAS meta-analysis.

6. It would be very interesting to confirm whether the SLC10A2 variants affect gallstone disease via reducing the transport activity. The authors could perform a mediation analysis to explore this possibility.

7. The section on data sharing does not mention sharing of the Icelandic RNA-seq expression data on blood and adipose that were used for the cis-eQTL analysis. How will these RNA-seq data be shared?

Reviewer #3 (Remarks to the Author):

In this report the authors compiled their GWAS results for gallstone disease (Nat Genet 2015) with data from the UK Biobank. They identified novel and replicated previously reported loci such as the bile acid transporter SLC10A2. Surprisingly the effects on systemic cholesterol concentrations showed no consistent changes with gallstone disease risk. The meta-analysis is innovative, carefully performed, and well written.

Comments:

1. p. 1, lines 1-2, abstract: The sentence appears to imply a need for prophylactic cholecystectomy to prevent acute pancreatitis, whereas in fact only symptomatic stones (biliary colic or stone complications such as acute cholecystitis or acute biliary pancreatitis) represent an indication for cholecystectomy. Please rephrase.

2. p. 2, line 13: ABCB4 is not only involved in but actually represents the hepatobiliary phosphatidylcholine (lecithin) transporter.

3. p. 4: Variants should be named consistently throughout the manuscript, e.g. it is not clear where variant ABCB4 Gly622Glu (p. 4) is listed in Supplementary Table 1, since the column variant (see Table 1) is missing in Supplementary Table 1.

4. pp. 5-6: The authors hypothesize that impaired bile acid reabsorption increases gallstone risk. However, decreased bile salt uptake in the ileum should result in increased hepatic bile acid

synthesis. Data on the bona fide bile salt synthesis marker 7 α -hydroxy-4-cholesten-3-one (C4) and FGF19 should be provided.

5. p. 6: Population attributable fractions (risks) should be calculated for the novel risk variants as well as composite genotypes.

6. p. 6, line 1: What is a "clear negative trend"? Data on correlation should be given (r ?, p ?).

7. p. 6, line 22: please double-check increased or decreased platelet count? Platelets decrease with progressive liver disease.

8. p. 8, FUT6: Does the recessive model suggested for gallstone disease also apply to vitamin B12 levels. Please discuss differences.

9. p. 8, FUT2: Please speculate how the stop-gained variant that increases susceptibility to GI infections could reduce gallstone risk.

10. p. 9, CD13: Is the functional consequence of the variant consistent with the promoter effect?

11. p. 11: The authors refer to ref. 40, which is apparently the first to indicate SLC10A2 as gallstone risk gene. The abstract states: "We have identified SLC10A2 as a novel susceptibility gene for cholelithiasis in humans. Comprehensive statistical analysis provides strong evidence that rs9514089 is a genetic determinant especially in male non-obese gallstone carriers. The minor allele of rs9514089 is related to differences in plasma cholesterol levels among the subjects." Please acknowledge this paper in more detail and state whether the two specific findings could be replicated.

12. p. 20, UK cohort: Self-reporting leads to a bias for symptomatic gallstone disease. This should be taken into account in calculating risks. Was there a difference between self-reported data and hospital records. On p. 7, the authors state that "nine of the UK individuals have a gallstone disease diagnosis". Please provide more specific information on these nine patients. Do they carry asymptomatic or symptomatic stones? Did they undergo cholecystectomy for symptomatic stones or other reasons?

13. p. 12: Discussion should include concepts how MZ heterozygosity could promote gallstone formation.

14. Table 2: please provide absolute number of individuals with respective genotypes

Minor edits:

- Correct references 24 and 25.

- p. 11, line: type 2 diabetes instead of type II

Point-by-point answer to reviewers

We would like to thank the three reviewers for their critique. We have made substantial changes to the manuscript in accordance with their comments and have attempted to address all of their concerns.

All changes to the originally submitted manuscript can be reviewed in track-changes. Line numbers below refer to edits in the manuscript with track changes using the “Simple Markup” option in Word 2016. References to relevant literature are made using PubMed identifiers (PMID).

For added clarity, we have marked initial unnumbered comments/questions by each reviewer by capital letters A, B, C, etc. In some cases, we have subdivided comments/questions from the reviewers, e.g. comment 1a), 1b), etc. Comments/questions from the reviewers are marked in *italic and bold font*, while our replies are in regular font.

Reviewer #1 (Remarks to the Author):

Overall:

Ferkingstad E and colleagues perform a comprehensive genome-wide association study for gallstone disease in deCODE and the UK Biobank, comprising of 27,174 cases and 736,838 controls, and the largest GWAS for this condition to date. Recently, Joshi AD et al Gastroenterology 2016 reported a GWAS for gallstones in 8,720 cases and 55,152 controls (discovery) and 6,489 cases and 62,797 (controls). They replicate these findings as well as from other studies and demonstrate new associations. The study was comprehensively and rigorously performed by an excellent research time.

COMMENT A: Beyond performing a larger GWAS, the authors do not convey new insights. The last sentence of Joshi AD et al’s abstract reads: “In this large-scale GWAS of gallstone disease, we identified 4 loci in genes that have putative functions in cholesterol metabolism and transport, and sulfenylation of bile acids or hydroxysteroids.” This paper’s abstract’s last sentence reads: “Taken together with replication of previously reported variants, this highlights the role of cholesterol metabolism and the enterohepatic

circulation of bile acids in gallstone disease.” The conclusions essentially are the same and highlights a major deficiency of the paper. While novel loci and variants are discovered, there are no novel insights.

ANSWER TO COMMENT A:

We acknowledge that we should highlight the novel insight gained from our study compared to previous findings more clearly. In the following points we list the novel contributions of our work, which we are now emphasizing in the text.

1. For the novel gallstone disease associated loci we observe an association at the loci of four out of the twelve genes linked to maturity onset diabetes of the young (MODY). Three of the novel gallstone associated variants are at loci of hepatocyte nuclear factor genes with mutations causing MODY (*HNFI1A*, *HNFI1B* and *HNFI4A*). In addition, we observe an association of *GCKR* which is a regulator of *GCK* that in turn has mutations causing MODY.
2. Two of the novel gallstone variants are in fucosyltransferase genes (*FUT2* and *FUT6*). We also observe that the *FUT2* variant rs601338 influences the risk of gastrointestinal infection and gallstone disease in opposite directions, an example of antagonistic pleiotropy. This suggests fucosylation influences gallstone disease susceptibility through interaction with the intestinal microbiota. *FUT6* and *FUT2* are also discussed in comments 8 and 9 of Reviewer 3.
3. The current study adds further evidence to previous findings by Joshi et al. (2016, PMID: 27094239) for the role of genes related to cholesterol or bile acid metabolism in gallstone disease. Joshi et al. implicated genes at four loci (*ABCG8*, *GCKR*, *CYP7A1*, and *SULT2A1*). Based on a much larger dataset our study highlights the role of additional genes at six novel loci in cholesterol metabolism (*FADS2*, *FUT2*, *HNFI1A*, *HNFI4A*, *JMJD1C* and *MARCH8*). Additionally, we systematically assess the effect of 203 serum lipid associated variants on gallstone disease risk, which to our knowledge has not been done before. We observe that there is an enrichment among lipid associating variants with gallstone risk but the direction of the effect is inconsistent, with some variants increasing gallstone risk associating with lower serum lipid levels while other associate with increased levels.
4. Regarding bile acid metabolism, with the identification of the *SLC10A2* locus we now specifically highlight the importance of intestinal bile acid reabsorption in the pathogenesis of gallstone disease.

To address point A we have changed the title, the discussion section and the end of the abstract as follows:

New title: Genome-wide association meta-analysis yields 20 new loci associated with gallstone disease

Abstract, page 1, lines 39-45:

"Additionally, we describe 17 novel common variants that associate with gallstone disease. Among those, two are in fucosyltransferase genes, four associate with maturity onset diabetes of the young, and six associate with lipid metabolism. We systematically assessed the effects of lipid associated variants with gallstone disease risk and found inconsistent direction of effect. With the discovery of gallstone-associated variants in *SLC10A2*, we highlight a role of the intestinal compartment of the enterohepatic circulation of bile acids in gallstone disease susceptibility."

Discussion, page 14, lines 352-364:

"Among the novel gallstone variants two are in fucosyltransferase genes (*FUT2* and *FUT6*). Fucosyltransferases mediate fucosylation which is an abundant posttranslational modification of glycosylated proteins and lipids. *FUT2* is responsible for the majority of fucosylation in the gastrointestinal tract (PMID: 15958416, PMID: 2910493). Interestingly, we observe opposing effects of the *FUT2* variant rs601338 on the risk of gallstone risk and susceptibility to gastrointestinal infection. This constitutes an example of antagonistic pleiotropy where a variant has opposing effects on the risk of two diseases. The increased risk of gastrointestinal infection observed for the wild-type secretor allele rs601338[G] is likely due to the presence of fucosylated glycans in the gastrointestinal tract that serve as adhesion sites for pathogens. *FUT2* may influence biliary disease progression through interaction with the biliary and intestinal microbiota (PMID: 26468308, PMID: 11578976). However, the mechanism through which the rs601338 variant influences the risk of gallstone disease is unclear, as the *FUT2* locus is highly pleiotropic and is the subject of balancing selection in humans (PMID: 11404338, PMID: 19487333). Further investigation is needed to explain the role of fucosylation in the pathogenesis of gallstone disease."

Discussion, page 16, lines 401-406 (changes underlined):

“In summary, we have discovered 21 novel gallstone disease variants in the largest gallstone disease GWAS to date. The associations presented emphasize the role of sequence variants in genes involved in cholesterol homeostasis and specifically highlight the intestinal compartment of the enterohepatic circulation of bile acids in the pathogenesis of gallstone disease. We conclude that sequence variants affecting the amount of cholesterol secreted into bile, or the cholesterol/bile acid ratio, are likely to cause gallstone formation.”

COMMENT B: As before, the authors demonstrate an enrichment of cholesterol-related genes. And, as before, the authors demonstrate that serum cholesterol is unlikely to be causal (Stender S et al. J Hepatol. 2013).

ANSWER TO COMMENT B:

To address this comment, we have added the new paragraph below to the discussion section:

Discussion, page 15-16, lines 387-400:

“While our data, as well as data from others (PMID: 22922093), do not support a direct path between serum cholesterol levels and gallstone formation, some aspects of cholesterol metabolism may contribute to gallstone disease. Biliary cholesterol super-saturation is considered a major factor in promoting cholesterol nucleation and gallstone formation (PMID: 17547709, 22898925, 11391534). Thus sequence variants affecting the amount of cholesterol secreted into bile, or the cholesterol/bile acid ratio, are likely to impact gallstone formation. Depending on the mechanism of action of the cholesterol regulating genes, the allele associated with gallstone risk may not consistently decrease or increase serum cholesterol levels. For example, increased activity of the sterol-transporter *ABCG5/8* that has a direct role in promoting biliary cholesterol secretion and in reducing intestinal absorption of dietary cholesterol is expected to increase the risk of gallstones while lowering circulating cholesterol. In contrast, reduced function of *CYP7A1*, the rate-limiting enzyme in the conversion of cholesterol to bile acids in the liver, likely predisposes to gallstone formation through a decrease in bile acid synthesis. It has been postulated that the increase in serum cholesterol levels that has been observed in carriers of *CYP7A1* gallstone risk variants may be mediated through down-regulation of hepatic LDL receptors (PMID: 29529257).”

COMMENT C: While the authors now demonstrate the association of low-frequency coding mutations at SLC10A2 are associated with gallstone disease, prior common variants were previously described in smaller studies.

ANSWER TO COMMENT C:

Due to its known biological function and role in the enterohepatic circulation of bile acids (i.e. its transport function in reabsorption of bile acids from the ileum), *SLC10A2* has previously been considered to be a candidate gene for gallstone disease. However, the previously reported association of the variant (rs9514089[C]; PMID: 19823678) with gallstone disease is clearly a false positive, as shown by the following four points which we are now incorporating into the text:

1. The association of rs9514089[C] with gallstone disease claimed by Renner et al (2009, PMID: 19823678), with OR = 2.04 (95% CI: 1.19-3.55) and a reported p-value of 7.7×10^{-3} do not survive Bonferroni correction accounting for testing of the 30 markers in their manuscript (P-value threshold = $0.05/30 = 1.7 \times 10^{-3}$). Sample size: N=240 cases and 255 controls.
2. A subsequent study (Tönjes et al, 2011, PMID: 22093174) with a comparable sample size (183 cases and 826 controls) did not report any association of the marker reported by Renner et al. (2009) with gallstone disease: OR = 0.83 (95% CI: 0.63-1.09) and P = 0.19. This estimate is inconsistent with that of point 1 above (p-value for test of heterogeneity: $P_{\text{het}} = 0.003$).
3. Previous large-sample gallstone GWAS such as Joshi et al. (2016, PMID: 27094239) did not report rs9514089[C] to associate with gallstones.
4. In the meta-analysis of Icelandic and UK data described in our paper (with 27,174 cases and 736,838 controls), we do not observe an association of rs9514089[C] with gallstones: OR = 0.991 (95% CI: 0.973-1.010), P = 0.36. This result is inconsistent with that of point 1 above (p-value for test of heterogeneity: $P_{\text{het}} = 0.007$).

To answer the reviewer's comment, we have added a new Supplementary Table 12 showing gallstone association results for rs9514089[C]. We have also inserted the following new text into the manuscript:

Discussion, page 13, lines 328-337:

“Because of its role in the enterohepatic circulation, a connection between *SLC10A2* and gallstone disease risk has been postulated. In a candidate gene study (PMID: 19823678) with 240 gallstone cases and 255 controls, the rs9514089[C] *SLC10A2* intron variant was reported to associate with an increased risk of gallstone disease (OR = 2.04 [1.19-3.55]; $P = 7.7 \times 10^{-3}$). The reported p-value does not survive multiple testing correction accounting for the 30 markers tested in their manuscript. A subsequent study (PMID: 22093174) found no association of rs9514089[C] with gallstone disease (OR = 0.83 [0.63-1.09]; $P = 0.19$), and this variant has not been reported to associate with gallstone disease by others (PMID: 25807286, PMID: 27094239, PMID: 26039129, PMID: 25920552). In our Icelandic and UK data (with 27,174 cases) we find no association between rs9514089 and gallstone disease risk (OR = 0.991 [0.973-1.010]; $P = 0.36$; Supplementary Table 12). Thus, we conclude that the claimed association of rs9514089 with gallstone disease is a false positive.”

COMMENT D: Instead of focusing on the long list of associations and biological plausibility, additional analyses to characterize causal pathways complementary to enrichment analyses would be helpful. For example, what aspects of cholesterol metabolism are relevant, particularly if serum cholesterol is not causal?

ANSWER TO COMMENT D:

See answer to Comment B, where we describe how we added new text to the Discussion (page 15-16, lines 387-400) in order to answer this question.

COMMENT E: Can the relationship with diabetes be further explored?

ANSWER TO COMMENT E:

To further explore the relationship of gallstone disease and genes linked to maturity onset diabetes of the young (MODY) we tested 12,809 variants in 12 genes linked to MODY for association with gallstone disease adding a new supplementary data table (Supplementary Data 10). We did not observe additional signals associating with gallstone disease. We note that the MODY genes *HNF1A*, *HNF1B* and *HNF4A* are all important hepatocyte transcription factors and that may partially explain their relationship with gallstone disease. The following new text was added to the manuscript:

Results, page 11, lines 297-301:

“To further explore the relationship of gallstone disease and genes linked to maturity onset diabetes of the young (MODY), we tested 12,809 variants in 12 genes linked to MODY (including 36 known pathogenic MODY variants) for association with gallstone disease; no additional signals associating with gallstone disease were observed (Supplementary Data 10).”

As suggested by the reviewer in Major comment 6, we have now also addressed this point by performing new analyses where we tested the 32 gallstone disease associated variants for differential risk between diabetics and non-diabetics. We found no significant differences in the effect of gallstone associated risk between diabetics and non-diabetics. We have added a new Supplementary Fig. 12 showing this, as well as adding the text below to the manuscript, in a new subsection “Diabetes and gallstone disease”:

Results, page 11, lines 294-297:

“The relationship between gallstone disease and diabetes is not clear (PMID: 20089496). To explore whether there is a difference in the effect of gallstone associated sequence variants between diabetics and non-diabetics, we tested whether the 32 gallstone disease associated variants confer different risk on diabetics and non-diabetics and found no significant differences (Supplementary Fig. 12).”

COMMENT F: What is the difference between gallstone variants that also associate with pancreatitis versus not?

ANSWER TO COMMENT F: The two variants, rs56398830 in *SLC10A2* and rs11887534 in *ABCG8*, that associate with pancreatitis are among the 32 gallstone associated variants with the strongest effects on gallstone disease, the only stronger variants are two of the rare *ABCB4* coding variants. *SLC10A2* and *ABCG8* encode transporter proteins that provide the primary mechanism of intestinal bile acid absorption/recycling (*SLC10A2*) or cholesterol secretion into bile (*ABCG8*). As expected, since pancreatitis is often a consequence of gallstone disease we observe a positive correlation between gallstone disease and pancreatitis effect sizes (Supplementary Fig. 3). We note that the sample size for pancreatitis (N = 2,843) is substantially smaller than for gallstone disease (N = 27,174). We speculate that with a

larger sample size for pancreatitis we would observe more pancreatitis associations among the gallstone associated variants. We have added the following text to the manuscript:

Results, page 5, lines 147-148:

“Among the 32 gallstone-associated variants, effects on gallstone disease and acute pancreatitis were positively correlated ($r = 0.81$ [0.69-0.92]; $P = 1.7 \times 10^{-8}$, Supplementary Fig. 3).”

Major 1 a): I am a bit confused why SLC10A2 is considered a novel locus for gallstones. Prior human genetic studies have implicated this gene for gallstones in other, albeit much smaller, studies, which the authors appropriately acknowledge.

Answer: See our reply to Comment C above, where we describe our new analyses and the changes we have made to the text. We conclude that the previously reported association of the *SLC10A2* variant rs9514089 with gallstone disease reported by Renner et al. (2009, PMID: 19823678) is a false positive.

Major 1 b): Are the authors distinguishing novelty based on variants or loci or both? The Manhattan plot (Figure 2) is confusing because it uses a mix of this terminology.

Answer: To make the definition of novelty more clear we have changed the text in the caption (changes underlined) and always use the term “loci”:

Caption of Figure 2, page 34:

“Manhattan plot for gallstone disease meta-analysis association results ($N_{\text{cases}} = 27,174$ and $N_{\text{controls}} = 736,838$). Variants are plotted by chromosomal position (x-axis) and $-\log_{10}$ P-values (y-axis). A chi-square test was used when testing for association. Green = Novel loci represented by low-frequency variants, red = novel loci represented by common variants, blue = reported gallstone disease loci.”

2. a) *Hypertriglyceridemia is co-morbid with gallstones as well as pancreatitis. Additionally, disruptive mutations in SLC10A2 have been linked to severe hypertriglyceridemia (Love MW et al. ATVB 2001).*

Answer: The claimed link between loss-of-function variants in *SLC10A2* and hypertriglyceridemia reported by Love et al. (2001, PMID: 11742882) is only supported by a single heterozygous individual in a screen of a total of 20 cases with familial hypertriglyceridemia and has not been further supported since the initial publication in 2001. We have serum triglyceride measurements for 119,624 Icelandic individuals. Based on these measurements, we defined individuals as hypertriglyceridemic if they had at least two measurements of serum triglycerides higher than 5 mmol/L, resulting in 1,599 hypertriglyceridemia cases. The *SLC10A2* missense variants Pro290Ser and Val98Ile (which was seen experimentally (PMID: 21649730) to have abolished and reduced function, respectively) do not associate with hyperglyceridemia (OR = 1.23 [0.74-2.03] and P = 0.42 for Pro290Ser; OR = 0.93 [0.75-1.14] and P = 0.48 for Val98Ile). Our data thus do not support the claim that loss-of-function variants in *SLC10A2* cause hypertriglyceridemia. To reflect this, we have changed the text as described below.

Results, pages 6-7, lines 175-180:

“Finally, it is worth noting that a truncating variant in *SLC10A2* (not found in our data) has been claimed to cause hypertriglyceridemia (PMID: 11742882). In our data the *SLC10A2* variants Pro290Ser and Val98Ile (which were shown experimentally (PMID: 21649730) to have abolished and reduced function, respectively) do not associate (P > 0.4 for both variants) with hypertriglyceridemia, defined as having at least two serum triglyceride measurements higher than 5 mmol/L (N = 1,599; Supplementary Data 5). Thus, our data do not support the claim that loss-of-function variants in *SLC10A2* cause hypertriglyceridemia.”

b) *Why were triglycerides not included in the lipids analyses?*

Answer: We thank the reviewer for pointing this out. To address the reviewer’s point we have added triglyceride association results (using 119,624 individuals) for the variants associating with gallstone disease in the present meta-analysis to Supplementary Data 3 and gallstone associations for variants reported to associate with triglyceride levels to Supplementary Data 9.

In the main text we have added the following changes (changes are underlined):

Results, page 4, lines 101-110:

“We tested the 32 gallstone disease variants for association with five other related diseases of the biliary system (acute pancreatitis [N=3,401], cholecystitis [N=3,565], fibrosis/cirrhosis of liver [N=1,044], cholestasis of pregnancy [N=2,615], and gallbladder cancer [N=382]), six liver biomarkers (alanine aminotransferase [ALT; N=172,086], aspartate aminotransferase [AST; N=164,467], gamma glutamyltransferase [GGT; N=156,692], alkaline phosphatase [ALP; N=154,097], albumin [N=92,163], and bilirubin [N=109,748]), and four lipid traits (high-density lipoprotein [HDL; N=136,736], low-density lipoprotein [LDL; N=126,220], total cholesterol [N=150,211]) and triglycerides [N = 119,624], resulting in a total of $32 \times (5+6+4) = 480$ tests that had to be accounted for in the assessment of significance (Supplementary Data 2 and Supplementary Data 3).”

Results, page 4, lines 117-119:

“Four variants associate with HDL cholesterol, six with LDL cholesterol, nine with total cholesterol and six with triglycerides (Supplementary Data 3).”

Results, pages 9-10, line 263 and lines 272-285:

“Lipid variants and gallstone disease.

Based on these results and because of the high cholesterol content of most gallstones, we tested 203 variants representing signals reported to associate with either HDL cholesterol, LDL cholesterol, total cholesterol or triglyceride serum levels for risk of gallstone disease (Supplementary Data 8). There is an excess of gallstone associated variants among reported HDL cholesterol, LDL cholesterol, total cholesterol and triglyceride associated variants: eight out of 83 HDL cholesterol variants ($P = 5.3 \times 10^{-19}$ based on a binomial test, see Methods), nine out of 65 LDL cholesterol variants ($P = 1.1 \times 10^{-22}$), fourteen out of 77 total cholesterol variants ($P = 2.5 \times 10^{-36}$) and five out of 49 triglyceride variants ($P = 1.2 \times 10^{-12}$) also associate with gallstone disease when all 203 tests are accounted for in the assessment of significance. In total, 23 of the 203 unique variants associating with one or more of HDL cholesterol, LDL cholesterol, total cholesterol and triglycerides also associate with gallstone disease. However, there is no consistency in the directions of the effects on serum lipids and gallstone disease risk (Fig. 4). This indicates that lipid serum levels are not by themselves

causative factors in gallstone formation, even though cholesterol metabolism appears to have an impact on gallstone risk.”

Major comment 3: For the gene enrichment analyses, how are genes identified/prioritized for non-coding analyses? Additional (brief) details would be helpful instead of only referring the reader to the ToppGene paper.

Answer: To address this we have added a sentence to the main text that now reads as follow (changes are underlined):

Results, page 10, lines 264-269:

“The biological functions of the 32 gallstone associated variants were assessed by gene enrichment analysis using ToppGene (PMID: 19465376). Variants were assigned to genes if they were located inside the gene or located within 5 kbs upstream and downstream from a gene, on the basis of mappings from previous GWAS studies or based on evidence from eQTL data (GTEx). In total 31 variants could be assigned to 27 genes that were included in the subsequent analysis (see Supplementary Data 8 for inclusion criteria).”

Major comment 4: The observation of showing enrichment with cholesterol-related variants but not having consistency with respect to effect direction for gallstone risk is interesting and has been observed, which the authors acknowledge. It likely suggests that these lipid values are not causal, as the authors mention. But, in the same sentence they also say “even though cholesterol metabolism has an impact on gallstone risk.” It would be helpful to further explore this. A major claim of this paper is demonstrating the importance of cholesterol metabolism. If cholesterol-related lab values do not appear to be causal, are there aspects of cholesterol metabolism that may be important/causal?

Answer: See answer to Comment B, where we describe the changes we made to the Discussion section (pages 15-16, lines 387-400) in order to answer this question. We have also changed Figure 4 (showing effects of lipids vs effects on gallstone disease for 203 variants reported to be associated with lipid levels): In addition to adding triglyceride levels (as suggested by the reviewer), to make the relationships clearer, we have now added gene names to Figure 4 and have made sure all panels in the new Figure 4 are on the same scale.

Major comment 5: a) Figure 3B does not clearly demonstrate a dose dependency in SLC10A2 transport activity with odds for gallstone disease. I would consider an alternative figure or move to the Supplement. The authors also make the claim in the Discussion that there is a dose-dependent relationship. In Figure 3B, I just see substantially elevated risk with nearly 0% enzymatic activity and a non-linear relationship for the rest of the variants.

Answer a): We have replaced the previous Figure 3B with an alternative figure that uses allelic effect instead of genotypic effect, which better demonstrates the dose dependency between gallstone risk and *SLC10A2* transport activity. In the new figure we show the allelic effect on gallstone disease for the four different missense variants Pro290Ser, Val98Ile, Val159Ile and Ser171Ala with experimentally determined transported activity as reported by Ho et al (2011, PMID: 21649730). The previous Figure 3B (with genotypic predicted values) has been moved to supplementary material (Supplementary Fig. 4).

In the main text we have added the following changes:

Results, page 6, lines 161-170:

“The abovementioned study (PMID: 21649730) reported transport activity for Pro290Ser and Val98Ile as well as two other ASBT/*SLC10A2* variants, Val159Ile (rs60380208[T]) and Ser171Ala (rs188096[A]). For each of these missense variants, we compared transport activity measurements to the risk of gallstone disease (Fig. 3b). Lower transport activity is correlated with greater risk of gallstone disease ($r = -0.99$ [-1.00, -0.66], $p = 8.2 \times 10^{-3}$ for OR vs. activity; $r = -0.99$ [-1.00, -0.77], $p = 5.2 \times 10^{-3}$ for log(OR) vs. activity). Additionally, we show transport activity compared to the genotypic risk of gallstone disease for different genotypes of Pro290Ser, Val98Ile and wild-type (Supplementary Fig. 4). Together these data indicate that the variants in *SLC10A2* affect gallstone risk through impairment of the bile acid reabsorption function of ASBT.”

b) Is gross SLC10A2 activity relevant or are disrupting particular domains more relevant?

Answer b): As we discuss in the main text, we postulate that the effect of variants in *SLC10A2* on gallstone risk is mediated through *SLC10A2* transport activity. This conclusion was drawn agnostically with regard to the domain structure as the identified variants happens to be experimentally characterized. However, it is also of note that both the Pro290Ser and

Val98Ile missense variants are within predicted transmembrane domains (Fig. 3a). To clarify the discussion of these points, we have made changes to text as described below (changes are underlined):

Results, page 5, lines 137-138:

“Pro290Ser is in transmembrane domain seven which is highly conserved within the SLC10 family of proteins, and results in a substitution of a non-polar proline with a polar serine (PMID: 16541252)”

Results, page 6, lines 158-160:

“Val98Ile is located in transmembrane domain two and has been shown by others to reduce the transport activity by 42%.”

Major comment 6: Given the epidemiological relationship with diabetes as well as implication of MODY genes, are there variants that have differential risk on gallstone disease among diabetics vs non-diabetics?

Answer: We have now addressed this point by performing new analyses where we tested the 32 gallstone disease associated variants for difference in risk conferred on diabetics versus non-diabetics. We found no significant differences in the effect of gallstone associated risk between diabetics and non-diabetics. We have added a new Supplementary Fig. 12 showing this, as well as adding new text as described below:

Results, page 11, lines 294-297:

“The relationship between gallstone disease and diabetes is not clear (PMID: 20089496). To explore whether there is a difference in the effect of gallstone associated sequence variants between diabetics and non-diabetics, we tested whether the 32 gallstone disease associated variants confer different risk on diabetics and non-diabetics and found no significant differences (Supplementary Fig. 12).”

Minor:

- 1. *In the Abstract's second sentence, it is stated: "Due to the threat of complications, such as acute pancreatitis, it is commonly treated with cholecystectomy." This is not true for asymptomatic, or incidental, gallstones.***

Answer: The sentence has been changed to "Due to the threat of complications, such as acute pancreatitis, symptomatic gallstones are commonly treated with cholecystectomy."

- 2. *SERPINA1 and HNF4A should be italicized in the Abstract. Please check the manuscript throughout to ensure all gene names are italicized.***

Answer: This has been corrected, and we have checked the manuscript to ensure that all gene names are italicized.

- 3. *Please clarify whether genotyping or sequencing strategy was performed with regards to gallstone case status.***

Answer: As we apply a population-based recruitment approach, genotyping and sequencing was not performed based on gallstone case status. To address the reviewer's point we have clarified this point in the Methods section:

Methods, page 24, lines 588-589:

"We note that selection of individuals for whole-genome sequencing and chip genotyping was not performed with regards to gallstone case status."

- 4. *Why is there a roughly 2-fold prevalence of gallstones in the UK Biobank vs deCODE? Perhaps a Supplementary Table with demographics and relevant comorbidities could be helpful.***

Answer: We have added a table (Supplementary Table 1) showing demographics and relevant comorbidities (diabetes and body mass index). We conclude that the main

reasons for the higher prevalence of gallstone disease in the UK data are the following:

1. Different recruitment practices between Iceland and the UK: the Icelandic set is more representative of the population as a whole, including a large proportion of the population.
2. The Icelanders are on average younger and leaner than UK individuals, and many of them are too young to have developed gallstone disease.

5. Suggest changing “HDL” to “HDL cholesterol” and “LDL” to “LDL cholesterol.”

Answer: This has been changed throughout the manuscript.

6. On page 4, what is meant by “unrelated missense variants?”

Answer: This has been changed to (change underlined) “distinct missense variants” (meaning variants with very low r^2 (squared correlation); i.e. variants that are not influenced by each other in conditional analyses).

Reviewer #2 (Remarks to the Author):

This is an interesting study by Ferkingstad et al. performing a genome-wide association meta-analysis for gallstone disease utilizing large cohorts from Iceland and UK (total $n=27,174$ cases and 736,838 controls). The study identified 21 novel variants at 20 loci for gallstone disease. While these results are of interest in further elucidating the genetic component of gallstone disease, I have some comments that would need clarification, especially regarding the diagnostic criteria, novelty of the study, and the details how multiple testing was corrected.

1. *It appears that the controls were not screened for gallstone disease. The authors should give a justification for this design and its potential effects on the results.*

Answer: Cases were identified based on self report or available health records (International Classification of Diseases (ICD) codes indicative of gallstone disease). It is possible that among the controls are individuals with asymptomatic gallstones or ones that have been diagnosed, but the diagnoses are not available to us. The presence of cases among the controls would be expected to dilute the true effect size of sequence variants and reduce power but would not cause false positive associations. Screening for gallstones on the population level is not feasible and this would not be preferable since the cases would instead be diluted by asymptomatic individuals. To clarify this, we have added the following sentence to the manuscript:

Methods, page 24, lines 588-589:

“We note that selection of individuals for whole-genome sequencing and chip genotyping was not performed with regards to gallstone case status.”

2. How many of the 18,417 UK cases were diagnosed based on only self-reporting? To assess the robustness of self-reported diagnosis of gallstone disease, it would be important to check if the GWAS association results for gallstone disease were similar in the UK participants when using only the self-reported cases versus using the cases with the ICD diagnoses made by the physicians at the hospitals?

Answer: 4,067 of the 18,417 UK cases were based on only self-reporting. We have now looked for a difference between association results for self-reported and ICD10-diagnosed cases. There is no difference between effect sizes in the two groups (ICD10 and self-reported) for 30 of 32 variants. We have added new text to the manuscript describing these results, a new Supplementary Fig. 13 showing the results, and a new Supplementary Table 11 describing the two variants for which there was a difference between association results.

Results, page 11-12, lines 304-312:

“Since 4,067 of the 18,417 UK cases were based only on self-reporting, we have looked for a difference between association results for self-reported and ICD10-diagnosed cases. There is no difference between effect sizes in the two groups (ICD10 and self-reported) for 30 of the 32 variants found. The two variants with a significant difference between ICD10-based and self-reported-based effects are the novel

gallstone disease associated variant *SERPINA1* Glu366Lys (PI Z; rs28929474[T]) (P-value for test for heterogeneity: $P_{\text{het}} = 2.2 \times 10^{-4}$) and the previously reported *ABCG8* Asp19His (rs11887534[C], $P_{\text{het}} = 1.6 \times 10^{-9}$) missense variants, which both showed a larger effect when using self-reported cases than ICD10 code cases. However, both of these variants associate with both ICD10-based and self-reported gallstone disease (Supplementary Fig. 13, Supplementary Table 11).”

3. *It is not clear why the authors selected the variant annotation based design to correct for multiple testing (ref 12). The rationale for this should be provided.*

Answer: The flat genome-wide significance P-value threshold of 5.0×10^{-8} that has become the standard for GWAS accounts only for one million tests. We use a weighted Bonferroni correction that accounts for all 42,907,111 variants tested in the meta-analysis, where the weighting scheme is based on prior probabilities of finding true positives in each variant class. We have published this method (Sveinbjornsson et al., 2016, PMID: 26854916), and have systematically used it since its publication. Each hypothesis is rejected if the P-value $< (0.05 * \text{weight}) / 42,907,111$; where the weights are based on estimated enrichment of true positive signals in each annotation class, as well as the number of tested variants in each annotation class.

In order to state this more clearly in the paper, the following text has been added:

Results, page 3, lines 84-89:

“The significance thresholds used were 2.0×10^{-7} for high-impact variants (including stop-gained, frameshift, splice acceptor or donor, N=11,465), 3.9×10^{-8} for moderate-impact variants (including missense, splice-region variants and in-frame indels, N=197,583), 3.6×10^{-9} for low-impact variants (including upstream and downstream variants, N=2,971,445) and 5.9×10^{-10} for lowest-impact variants (including intron and intergenic variants, N=39,726,619).”

Methods, page 26, lines 641-647:

“We have previously demonstrated that the variant annotation based multiple testing method provides greater power to detect associations than Bonferroni correction where all variants are treated equally, while maintaining an overall family-wise error

rate of 0.05. All variants except loss-of-function variants (including stop-gained, frameshift, splice acceptor or donor) have p-value thresholds stricter than the common threshold of 5.0×10^{-8} , while loss-of-function variants were given a threshold of 2.0×10^{-7} due to their higher prior probability of representing true positive signals.”

4. *The authors describe results from a recessive model for SNP rs708686, upstream of FUT6. How many statistical models were tested all in all? Were the presented results also corrected for the number of these tested models?*

Answer: Only two models were tested: additive and recessive. Presented results were not corrected for the number of tested models. Correcting for the number of tested models (i.e. multiplying all reported GWAS p-values by two) would not change the list of reported variants in this manuscript.

5. *Previous studies have shown that variant Pro290Ser abolishes the bile acid transport activity of ASBT. It would be important for the reader that the authors would explain throughout the manuscript which results are new and which support previous data. Functional studies with the novel variants would expand the results beyond a GWAS meta-analysis.*

Answer: The association of missense variants in *SLC10A2* with gallstone disease reported in the current study is novel (see also answer to Reviewer 1, Major comment 1a). The assessment of transport activity of *SLC10A2* missense variants has been reported by Ho et al. (2011, PMID: 21649730). We compare the gallstone disease risk of the *SLC10A2* missense variants with their measured transport activity, and conclude that gallstone disease risk is negatively correlated with transport activity among the four variants which is a novel result. The functional work performed by Ho et al. (2001) is helpful to explain the functional impact of our reported *SLC10A2* variants. Further functional studies on the remaining variants are beyond the scope of the paper.

To further clarify which results are new and which support previous data we have added the following to the main text (changes are underlined):

Results, page 5, lines 135-136:

“Others have experimentally shown that Pro290Ser nearly abolishes the bile acid transport activity of ASBT (PMID: 7592981, PMID: 21649730) (< 3% of the wild-type level (PMID: 21649730)).”

Results, page 6, lines 158-160:

“Val98Ile is located in transmembrane domain two and has been shown experimentally by others to reduce the transport activity by 42% (PMID: 21649730)”

Results, page 6, lines 161-164:

“The abovementioned study (PMID: 21649730) reported transport activity for Pro290Ser and Val98Ile as well as two other ASBT/SLC10A2 variants, Val159Ile (rs60380208[T]) and Ser171Ala (rs188096[A]). For each of these missense variants, we compared the previously reported transport activity measurements to their effect on gallstone disease risk from the current study (Fig. 3b).”

6. *It would be very interesting to confirm whether the SLC10A2 variants affect gallstone disease via reducing the transport activity. The authors could perform a mediation analysis to explore this possibility.*

Answer: We have addressed this question by adding a new figure (Figure 3a) showing experimentally assessed transport activity from Ho et al (2011, PMID: 21649730) vs gallstone disease OR (effect sizes) from our meta-analysis of Icelandic and UK data for four different missense variants in *SLC10A2*, including Pro290Ser and Val98Ile that were found to be associated with gallstone disease at a genome-wide significant level. The new Figure 3a hopefully makes the relationship between *SLC10A2* transport activity and gallstone disease risk clearer.

Also see answer to Reviewer 2, Major comment 5.

7. *The section on data sharing does not mention sharing of the Icelandic RNA-seq expression data on blood and adipose that were used for the cis-eQTL analysis. How will these RNA-seq data be shared?*

Answer: The hypothesis was generated based on publicly available data from the Gene Tissue Expression (GTEx) database. The Icelandic data were used for the purpose of confirming those results. We have removed all references to the Icelandic RNA expression data in the manuscript. Changes to the text are described below.

Methods:

The subsection on “RNA sequencing analysis” has been removed, since we no longer use these data.

Results, page 11, lines 290-291:

Discussion of RNA-seq data has been removed in the sentence that now reads “We replicate the association of rs1880118[C] with increased HDL cholesterol in our data (effect = 0.15 SD [0.10-0.20], $P = 2.4 \times 10^{-9}$).”

Supplementary Table 2:

The rows of this table that referred to Icelandic RNA-seq expression data have been removed.

Reviewer #3 (Remarks to the Author):

In this report the authors compiled their GWAS results for gallstone disease (Nat Genet 2015) with data from the UK Biobank. They identified novel and replicated previously reported loci such as the bile acid transporter SLC10A2. Surprisingly the effects on systemic cholesterol concentrations showed no consistent changes with gallstone disease risk. The meta-analysis is innovative, carefully performed, and well written.

Comments:

- 1. p. 1, lines 1-2, abstract: *The sentence appears to imply a need for prophylactic cholecystectomy to prevent acute pancreatitis, whereas in fact only symptomatic stones (biliary colic or stone complications such as acute cholecystitis or acute biliary pancreatitis) represent an indication for cholecystectomy. Please rephrase.***

Answer: The sentence has been changed to (changes to the text are underlined) “Due to the threat of complications, such as acute pancreatitis, symptomatic gallstones are commonly treated with cholecystectomy.” (This was also pointed out by Reviewer 1, Minor comment 1.)

- 2. p. 2, line 13: *ABCB4 is not only involved in but actually represents the hepatobiliary phosphatidylcholine (lecithin) transporter.***

Answer: We have now changed the sentence, which reads as follows (changes are underlined):

Introduction, page 2, lines 64-65:

“*ABCB4* encodes the ABC transporter responsible for the secretion of lecithin into bile.”

- 3. p. 4: *Variants should be named consistently throughout the manuscript, e.g. it is not clear where variant *ABCB4 Gly622Glu* (p. 4) is listed in Supplementary Table 1, since the column variant (see Table 1) is missing in Supplementary Table 1.***

Answer: We have added rs-names of the markers to the text where they were missing, so they can be found in the tables.

- 4. p. 5-6: *The authors hypothesize that impaired bile acid reabsorption increases gallstone risk. However, decreased bile salt uptake in the ileum should result in increased hepatic bile acid synthesis. Data on the bona fide bile salt synthesis marker 7 α -hydroxy-4-cholesten-3-one (C4) and FGF19 should be provided.***

Answer: We do not have measurements of 7 α -hydroxy-4-cholesten-3-one (C4). For FGF19, we have measurements for 294 individuals, of which four are carriers of the Pro290Ser *SLC10A2* variant (all heterozygotes). Note that these are the same four Pro290Ser carriers for which we also have serum taurocholate measurements. There was no significant effect of Pro290Ser on FGF19 levels. We have added a new Supplementary Table 5 providing taurocholate and FGF19 measurements for the four Pro290Ser carriers that had available measurements, and added new text to the manuscript as detailed below.

Results, pages 6, lines 152-154:

“We also tested for an effect of Pro290Ser on fibroblast growth factor 19 (FGF19, N = 293), a hormone that regulates bile acid synthesis (PMID: 17072310), and observed no effect (Effect: -0.08 SD [-1.02, 0.87]; P = 0.87; Supplementary Table 5).”

5. p. 6: Population attributable fractions (risks) should be calculated for the novel risk variants as well as composite genotypes.

Answer: To address the question we have added calculations of population attributable fractions (PAF) to supplementary Data 4. We note that PAF is a measure of the impact of a sequence variant on a disease from a public health point of view. It is however a problematic measurement that is not the right one to evaluate the contribution of sequence variants to disease risk (PMID: 25223781). For that purpose, the sibling recurrent risk ratio is more appropriate.

The following sub-section has been added to the Methods:

Methods, page 27, lines 666-674:

“Population attributable fraction

Population attributable fraction is defined as the fraction of cases that would be eliminated from the population if the risks of all individuals carrying the risk variant could be contained, e.g. through a treatment, to be the same as non-carriers of the at-risk variant(s). It can be calculated for a variant using the following formula:

$$\text{PAF} = 1 - (1/W) \text{ where } W = (1 - p)^2 + 2p(1 - p)\text{RR} + p^2\text{RR}^2$$

Here 'p' denotes the at-risk allele frequency and RR is the relative risk of for a disease. RR are estimated by odds ratios under the assumption of the multiplicative (that is, logarithmic additive) model, so the RR for carrying two risk variants is RR^2 ."

6. *p. 6, line 1: What is a "clear negative trend"? Data on correlation should be given (r?, p?).*

Answer: Data on correlation has been added; see changes to the text detailed below.

Results, page 6, lines 164-166:

"Lower transport activity is correlated with greater risk of gallstone disease ($r = -0.99$ [-1.00, -0.66], $p = 8.2 \times 10^{-3}$ for OR vs. activity; $r = -0.99$ [-1.00, -0.77], $p = 5.2 \times 10^{-3}$ for log(OR) vs. activity)".

Also note that we have replaced Figure 3B with an alternative figure that shows more clearly the dose dependency. In the new figure we show the allelic effect on gallstone disease for the four different missense variants Pro290Ser, Val98Ile, Val159Ile and Ser171Ala together with experimentally determined transported activity as reported by Ho et al (2011, PMID: 21649730). The previous Figure 3B has been moved to supplementary material (Supplementary Fig. 4).

7. *p. 6, line 22: please double-check increased or decreased platelet count? Platelets decrease wit progressive liver disease.*

Answer: We have double-checked this. The rs28929474[T] allele indeed associates with increased platelet counts in the general population. We note that the large majority of individuals with platelets count measurements are disease free.

8. *p. 8, FUT6: Does the recessive model suggested for gallstone disease also apply to vitamin B12 levels. Please discuss differences.*

Answer: We now have calculated genotypic effects on vitamin B12 levels (based on Icelandic data) for the different *FUT6* rs708686 genotypes, see the new Supplementary Figure 13. While there is an increased effect on vitamin B12 in C/T heterozygotes, the effect on T/T homozygotes is greater than expected from an additive model compared to C/C homozygotes. We have added text to the manuscript describing this (additions are underlined):

Results, page 9, lines 229-232:

“We have previously reported rs708686[T] to associate with increased levels of vitamin B12 (PMID: 23754956) (Supplementary Data 6). Similarly with what we see for gallstone disease, we observed a trend towards recessive association of rs708686 with vitamin B12 levels (Supplementary Fig. 11).”

9. p. 8, *FUT2*: Please speculate how the stop-gained variant that increases susceptibility to GI infections could reduce gallstone risk.

Answer: Note that rs601338[A] corresponds to the stop-gained variant. The wild-type secretor allele (rs601338[G]) that associates with reduced gallstone risk in the current study has been reported to associate with susceptibility to GI infection by norovirus and *Helicobacter pylori* (PMID: 16306606, PMID: 11535550). *FUT2* is one of the most pleiotropic loci in the literature (EBI gwas catalog) and rs601338, and highly correlated variants ($r^2 > 0.9$), have been reported by us and others to have an influence on several disease and quantitative traits (PMID: 29691392, PMID: 23754956, PMID: 29875488). Consistent with previous observations, rs601338[G] associates with increased risk of gastrointestinal infections (ICD-10 code A08*) in our data (OR = 1.6, $P = 7 \times 10^{-9}$) and in the UK (OR = 1.2, $P = 0.01$) (see the new Supplementary Table 11). The effect of rs601338, in opposite directions, on gallstones and GI infection could be mediated by different mechanisms since *FUT2* acts as a post-translational modifier of several glycoproteins. However, it has been speculated that *FUT2* might influence biliary disease progression through interaction with the biliary and intestinal microbiota (PMID: 26468308). In support of this, we note that *FUT2* is highly expressed in the digestive tract including the small intestine) and is also at high level of expression in the gallbladder (according to the human protein atlas database, <https://www.proteinatlas.org/>).

In line with previous reports we observe associations ($P < 2.0 \times 10^{-7}$) of rs601338[G] with a range of quantitative traits including associations with decreased vitamin B12, increased alkaline phosphatase, decreased lipase, decreased tumor antigens (CA 19-9, CEA and CA 125), decreased total cholesterol and increased FAM3D protein expression (Supplementary Table 11). In addition, we observe associations with decreased gamma glutamyl transferase, increased folate and increased uric acid, which to our knowledge has not been previously reported (Supplementary Table 11). To address the reviewers question we have modified the text as described below (changes underlined):

Results, page 9, lines 236-244:

“The wild-type secretor allele rs601338[G] has been implicated in susceptibility to gastrointestinal infections (PMID: 16306606, PMID: 11535550), decreased vitamin B12 levels (PMID: 18776911) and with effects on several diseases and other traits (PMID: 29691392) (Supplementary Data 6). Consistent with previous observations, rs601338[G] associates with increased risk of gastrointestinal infections (ICD-10 code A08*) in our data (OR = 1.56 [1.34-1.82], $P = 7.2 \times 10^{-9}$) (Supplementary Table 10). In Iceland we also observe an effect of rs601338[G] on decreased gamma glutamyl transferase and increased folate (Supplementary Table 10). *FUT2* is a paralog of the previously mentioned *FUT6* and they both belong to the gene family of fucosyltransferases (63% identical on the protein level).”

Discussion, page 14, lines 352-364:

“Among the novel gallstone variants, two are in fucosyltransferase genes (*FUT2* and *FUT6*). Fucosyltransferases mediate fucosylation which is an abundant posttranslational modification of glycosylated proteins and lipids. *FUT2* is responsible for the majority of fucosylation in the gastrointestinal tract (PMID: 15958416, PMID: 2910493). Interestingly, we observe opposing effects of the *FUT2* variant rs601338 on the risk of gallstone disease and susceptibility to gastrointestinal infection. This constitutes an example of antagonistic pleiotropy where a variant has opposing effects on the risk of two diseases. The increased risk of gastrointestinal infection observed for the wild-type secretor allele rs601338[G] is likely due to the presence of fucosylated glycans in the gastrointestinal tract that serve as adhesion sites for

pathogens. It has been speculated that *FUT2* might influence biliary disease progression through interaction with the biliary and intestinal microbiota (PMID: 26468308, PMID: 11578976). However, the mechanism through which the rs601338 variant influences the risk of gallstone disease is unclear, as the *FUT2* locus is highly pleiotropic and is the subject of balancing selection in humans (PMID: 11404338, PMID: 19487333). Further investigation is needed to explain the role of fucosylation in the pathogenesis of gallstone disease.”

The following footnote has been added to Table 1:

“***the tested allele rs601338[G] corresponds to the classical wild-type secretor allele of *FUT2*”

10. p. 9, *CD13*: Is the functional consequence of the variant consistent with the promoter effect?

Answer: As we point out in the main text, *CD13* (alias *ANPEP*) has been reported to promote cholesterol crystallization in a in-vitro system (PMID: 7907074). We do not know the effect of the missense variant Ala311Val (rs17240268[A]) on *CD13* protein function. However, we expect rs17240268[A] to have a reducing effect on protein function for the observed protective effect of the minor allele rs17240268[A] on gallstone disease to be consistent with the reported promoting effect of *CD13* on cholesterol crystallization.

To address this point we have added the following text:

Results, page 9, lines 248-250:

“The minor allele rs17240268[A] would be expected to have a reducing effect on protein function for the observed protective effect on gallstone disease to be consistent with the reported promoting effect of *ANPEP* on cholesterol crystallization.”

11. p. 11: The authors refer to ref. 40, which is apparently the first to indicate *SLC10A2* as gallstone risk gene. The abstract states: "We have identified *SLC10A2* as a novel susceptibility gene for cholelithiasis in humans. Comprehensive statistical analysis provides strong evidence that rs9514089 is a genetic determinant

especially in male non-obese gallstone carriers. The minor allele of rs9514089 is related to differences in plasma cholesterol levels among the subjects." Please acknowledge this paper in more detail and state whether the two specific findings could be replicated.

Answer: See our reply to Comment C by Reviewer 1 above, where we describe our new analyses and the changes we have made to the text, and where we conclude that the previously reported association of the *SLC10A2* variant rs9514089 with gallstone disease reported by Renner et al. (2009, PMID: 19823678) is a false positive.

12.

a. p. 20, UK cohort: Self-reporting leads to a bias for symptomatic gallstone disease. This should be taken into account in calculating risks. Was there a difference between self-reported data and hospital records.

Answer: See our answer to Reviewer 2's comment 1, where we describe how we made new analyses and changes to the text in order to answer this question.

b. On p. 7, the authors state that "nine of the UK individuals have a gallstone disease diagnosis". Please provide more specific information on these none patients. Do they carry asymptomatic or symptomatic stones? Did they undergo cholecystectomy for symptomatic stones or other reasons?

Answer: All nine PI ZZ homozygotes (six females and three males) were classified as gallstone disease cases based on ICD codes diagnosed between 44 and 66 years of age. Five out of the nine individuals have cholecystitis. All but one have undergone cholecystectomy indicating symptomatic gallstone disease. We have added a new Supplementary Table 8 with ICD and OPCS procedure codes indicative of gallstone disease assigned to UK PI ZZ carriers.

The text has been changed as follows (changes are underlined):

Results, page 7, lines 199-201:

"...nine of the UK individuals have a gallstone disease diagnosis, and all but one have undergone cholecystectomy indicating symptomatic disease (Supplementary Table 8)"

13. p. 12: Discussion should include concepts how MZ heterozygosity could promote gallstone formation.

Answer: To address this point we have added the following text to the discussion:
Discussion, page 14-15, lines 365-373:

“We show that the PI Z allele of *SERPINA1* associates with increased risk of gallstone disease. Severe AAT deficiency in PI ZZ homozygotes predisposes to emphysema and, less commonly, liver disease (PMID: 21960536). The precise pathomechanism of liver disease associated with severe AAT deficiency is not well understood but histopathological findings of liver parenchyma in PI ZZ homozygotes indicate that liver abnormalities result from toxic intracellular accumulation of AAT in hepatocytes (PMID: 15619203). Moderate AAT deficiency in PI MZ heterozygotes has been associated with slightly increased risk of liver disease and intracellular accumulation of AAT in hepatocytes (PMID: 24661570). We speculate that MZ heterozygosity may result in general liver dysfunction as result of toxic accumulation of AAT in hepatocytes over time, thus predisposing to gallstone disease.”

14. Table 2: please provide absolute number of individuals with respective genotypes

Answer: This information has been added to Table 2.

Minor edits:

- Correct references 24 and 25.

Answer: This has been corrected.

- p. 11, line: type 2 diabetes instead of type II

Answer: This has been corrected.

Reviewer #1 (Remarks to the Author):

The authors present a substantially revised manuscript. They were very thorough in responding to all of my concerns and questions; they have addressed these adequately.

Reviewer #2 (Remarks to the Author):

The authors have successfully addressed my comments. I have no further comments.

Reviewer #3 (Remarks to the Author):

Point-by-point-answers, p. 22:

The authors have added „Population attributable“ (PAF) to the Methods section, but refer to Suppl. Data 4, which now lists sibling recurrent risk ratios.

Here, PAF should be added, too, and sibling recurrent risk ratios should be included in Methods.

Replies to reviewers' comments for Ferkingstad et al., "Genome-wide association meta-analysis yields 20 loci associated with gallstone disease"

Our replies are in bold font.

REVIEWERS' COMMENTS:

Reviewer #1 (Remarks to the Author):

The authors present a substantially revised manuscript. They were very thorough in responding to all of my concerns and questions; they have addressed these adequately.

Reviewer #2 (Remarks to the Author):

The authors have successfully addressed my comments. I have no further comments.

Reviewer #3 (Remarks to the Author):

Point-by-point-answers, p. 22:

The authors have added „Population attributable“ (PAF) to the Methods section, but refer to Suppl. Data 4, which now lists sibling recurrent risk ratios.

Here, PAF should be added, too, and sibling recurrent risk ratios should be included in Methods.

Response: PAF has been added to Suppl. Data 4. Sibling recurrent risk ratios are included in Methods.